# Naturally-associated bacteria modulate Orsay virus infection of *Caenorhabditis elegans*

**Rubén González****\*, Marie-Anne Félix\***

Institut de Biologie de l'École Normale Supérieure, CNRS, INSERM, Paris, France

\* ruben.gonzalez@bio.ens.psl.eu (RG); felix@bio.ens.psl.eu (M-AF)

## Abstract

Microbes associated with an organism can significantly modulate its susceptibility to viral infections, but our understanding of the influence of individual microbes remains limited. The nematode *Caenorhabditis elegans* is a model organism that in nature inhabits environments rich in bacteria. Here, we examine the impact of 71 naturally associated bacteria on *C. elegans* susceptibility to its only known natural virus, the Orsay virus. Our findings reveal that viral infection of *C. elegans* is significantly influenced by monobacterial environments. Compared to an *Escherichia coli* environmental reference, the majority of tested bacteria reduced *C. elegans* susceptibility to viral infection. This reduction is not caused by virion degradation or poor animal nutrition by the bacteria. The repression of viral infection by the bacterial strains *Chryseobacterium* JUb44 and *Sphingobacterium* BIGb0172 does not require the RIG-I homolog DRH-1, which is known to activate antiviral responses such as RNA interference and transcriptional regulation. Our research highlights the necessity of considering natural biotic environments in viral infection studies and opens the way future research on host-microbe-virus interactions.

## Author summary

*Caenorhabditis elegans* is a nematode roundworm that naturally inhabits decomposing vegetal matter—a bacteria-rich environment in which the animal feeds. The sampling of *C. elegans* in its natural habitats led to the discovery of its only known natural virus, the Orsay virus. This finding has potentiated virology research using *C. elegans*. However, most studies have been conducted using a laboratory bacterial environment. Here, we studied the effect of bacteria associated in nature with the nematodes on their susceptibility to Orsay virus infection. We tested a diverse set of bacteria, most of which come from regions where the Orsay virus is present. We found that most of these natural bacteria reduce susceptibility to infection compared to the reference laboratory bacterial environment. We studied some of the bacteria in depth and found that some can induce resistance to infection even in animals lacking the main antiviral immune responses. Our study highlights the key role that associated microbes play in host-virus

**Data Availability Statement:** All relevant data are within the manuscript and its Supporting information files.

**Funding:** R.G. is funded by an EMBO Postdoctoral Fellowship (ALTF 311-2021). M.A.F. is funded by

the Centre National de la Recherche Scientifique. This work was partially funded through a grant from the Agence Nationale de la Recherche ANR-19-CE12-0025. The funders had no role in study design, data collection and analysis, decision to publish, or preparation of the manuscript.

**Competing interests:** The authors have declared that no competing interests exist.

interactions and underscores the importance of considering natural environments in research.

## Introduction

The biotic environment of an organism influences many of its traits, including its immune responses [1]. In particular, microbes can influence interactions with pathogenic viruses [2]. In some cases, commensal and pathogenic microbes can enhance their host immunity and reduce viral replication and infectivity, thus providing protection against infection [3–6]. Conversely, some microbes can increase their host's susceptibility to viral infections; some of them might even be crucial for new viruses to successfully infect their hosts [7–9]. To these intricate three-way interactions between host, virus, and microbes must be added the complex relationships among the individual microbes that form the associated microbial community. The study of simplified pathosystems and associated microbes would facilitate the understanding of these interactions and of host immunity to viral infections.

The nematode *Caenorhabditis elegans*, widely used as a model organism [10], presents an ideal model system to study microbe–host interactions, which enabled insightful discoveries over the past 20 years [11]. One key feature of *C. elegans* for such studies is that it is possible to free the animals of associated microbes using a bleach treatment to which the embryos resist, and then reassociate them at will with one or several microbes.

The natural habitat of *C. elegans* is rotting fruit or other decomposing vegetal matter, which is rich in bacteria. These bacteria not only serve as a primary food source for the nematode but also constitute part of its biotic environment [12,13]. In the last decade, bacteria associated with *C. elegans* in its natural habitats have been characterized, revealing their presence in the nematode's external environment, intestinal lumen, and as a fraction that is lysed and digested as food [14–16]. In samples taken from Europe (specifically France and Spain), it was found that the most abundant phylum was *Pseudomonadota* (synonym: *Proteobacteria*, a group of Gram-negative bacteria). Other phyla such as *Bacteroidota* (synonym: *Bacteroidetes*, Gram-negative bacteria), *Bacillota* (synonym: *Firmicutes*, Gram-positive bacteria), and *Actinomycetota* (synonym: *Actinobacteria*, Gram-positive bacteria) were also identified. Furthermore, apple samples rich in *Proteobacteria* were observed to promote the proliferation of *C. elegans* [16]. Bacterial environments have been shown to affect *C. elegans* metabolism, signaling pathways, and immune responses [17–20]. While associated bacteria can protect *C. elegans* from extracellular pathogens such as bacteria or fungi [21–26], their effect in modulating interactions with intracellular pathogens, including viruses, remains unexplored.

The only known naturally occurring virus in *C. elegans* is the Orsay virus (OrV) [27]. The virus was first isolated from animals found on a rotting apple in Orsay, France [27]. OrV is a positive single-stranded RNA virus with a bipartite genome similar to that of Nodaviruses. This virus enters new hosts when *C. elegans* ingests the virions while feeding and, once inside the host intestinal lumen, it infects the intestinal cells. Infected intestinal cells release virions that are further transmitted horizontally to other individuals through the fecal-oral route. OrV infection affects host fitness by reducing and delaying reproduction [28]. *C. elegans* lacks immune components found in other animals, but is able to mount an immune response against viral infection [29]. Known mechanisms include the use of RNA interference (RNAi) and uridylation responses to target viral RNA for degradation, ubiquitin-mediated pathways that may target viral proteins for degradation, and transcriptional activation of specific genes in response to infection. This transcriptional response includes a specific immune response to

*C. elegans*'s intestinal intracellular pathogens (i.e., microsporidia or viruses) known as the Intracellular Pathogen Response (IPR) [30]. The IPR involves the upregulation of a limited number of genes, including genes belonging to the *pals* gene family (whose biochemical function is currently unknown) and genes predicted to encode ubiquitin ligase components. The IPR is dependent on the helicase DRH-1, a RIG-I family member, which is likely a viral sensor and is also essential for the RNAi response. Downstream of DRH-1, the IPR is partially dependent on the transcription factor ZIP-1 [31–33]. Severe infections trigger a general 'biotic stress response', which involves the upregulation of stress response genes (such as *lys-3*) [34]. Many mechanisms and components involved in the *C. elegans* immune response to Orsay virus are evolutionary conserved [31,32,34,35]. Therefore, studying these responses can provide valuable insights into virus-host interactions in other organisms.

In this study, we investigated the effect of bacterial environments on *C. elegans* susceptibility to OrV. We focused on monocultures of bacterial clones isolated from *C. elegans* natural habitats [14–16,36–38], which for short will be referred to as "natural bacteria", many of which were isolated in locations where the Orsay virus was also found. We primarily investigated bacterial strains from the phylum *Pseudomonadota*. Additionally, we studied bacterial strains from *Bacillota*, *Bacteroidota*, and *Actinomycetota*. Our objective was to determine whether the bacterial environment affects the nematode's response to the virus and gain a deeper understanding of how specific bacterial strains may modulate host susceptibility to viruses. By investigating the interplay between *C. elegans*, its natural bacterial environment, and the Orsay virus, our study aims to provide novel insights into the role of the microbial context in modulating host-pathogen interactions and serve as a foundation for further studies with the potential to discover previously unknown viral defense mechanisms.

## Results

### Single bacterial environments modulate *C. elegans* susceptibility to viral infection

We examined the impact of 71 natural bacterial strains (S1 Table) on the susceptibility of *C. elegans* to OrV. For this initial screen, we used animals carrying the *pals-5p*::*GFP* reporter of intracellular infection [39]. We conducted the screen in five experimental blocks (S1 Fig), testing three populations of ≈100 animals per bacterial environment and including *Escherichia coli* OP50 in each block as the reference. We transferred axenic embryos to plates seeded with a single bacterial strain, inoculated the plate with OrV, and 72 hours post inoculation (hpi) scored the proportion of animals activating the *pals-5* reporter with and without viral exposure. None of the bacteria triggered the *pals-5* reporter activation in the absence of virus (S2 Fig). Based on the relative proportion of virus-inoculated animals in the population showing *pals-5p*::*GFP* activation compared to virus-inoculated populations on the *E. coli* OP50 reference (Fig 1, upper panel), we categorized the bacterial strains in three groups: 59 repressors of *pals-5* reporter activation (significant lower activation than OP50), 5 enhancers (significant higher activation than OP50), and 7 bacteria with an effect similar to that of the *E. coli* environment. We tested some bacterial strains across different blocks, verifying that their effect on viral infections was reproducible (S3A Fig); all 6 bacteria with a relative rate higher than 1 were examined again within the same block, successfully replicating the observations of the initial screen (S3B Fig). For the bacteria in which no reporter activation was observed upon viral infection, we wondered whether the bacterial environments prevented *pals-5* reporter activation under other types of stressing conditions, such as heat stress [30]. The *pals-5* reporter could be activated by heat stress in these bacterial environments, showing that the suppression was specific to the response to viral infection (S1 Table).

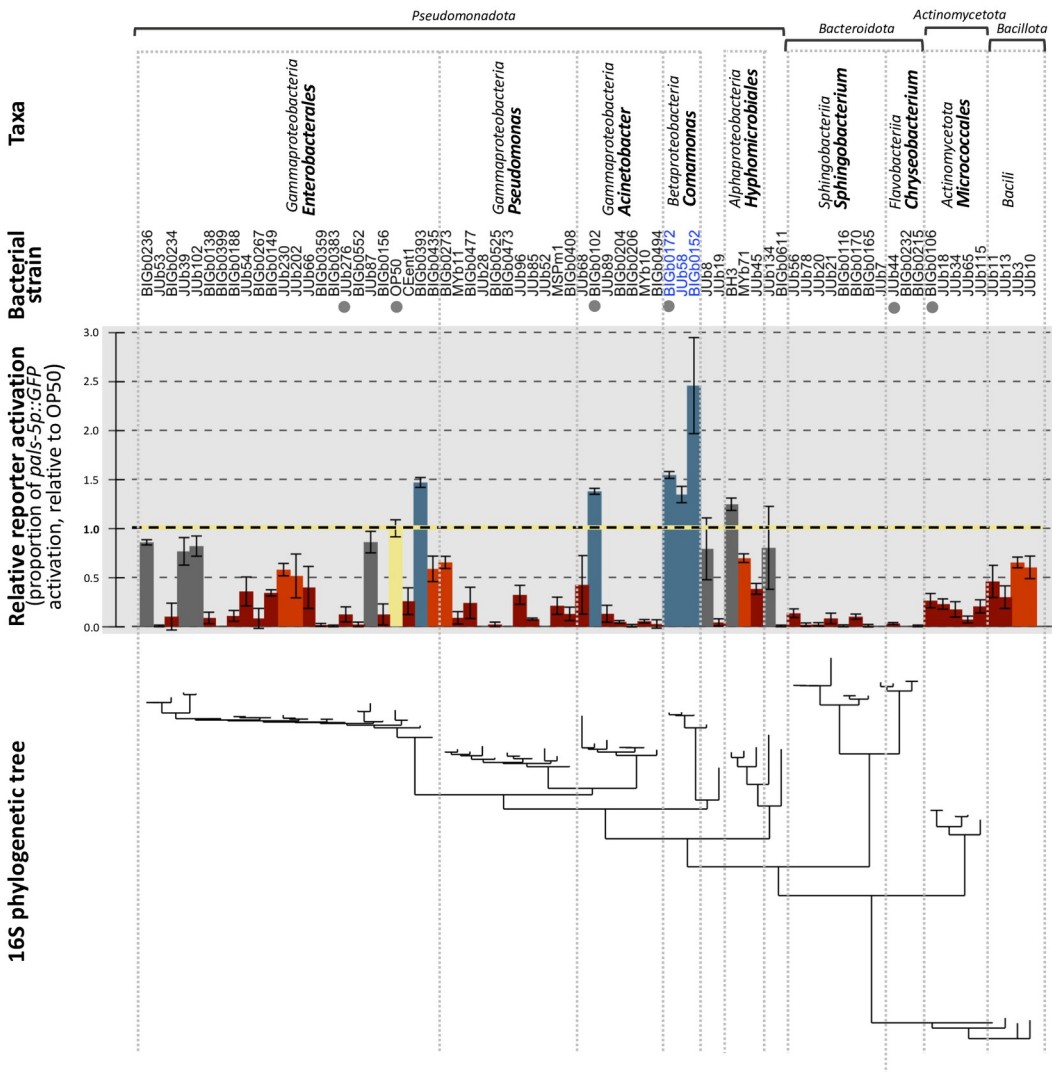

**Fig 1. Most naturally associated bacteria reduce *C. elegans* susceptibility to viral infection compared to *E. coli* OP50.** For each bacterial environment, three replicates of ca. 100 ERT54 animals were challenged with the Orsay virus JUv1580. The proportion of animals activating the *pals-5p*::*GFP* reporter was measured at 72 hpi (raw data are in S2 File). Data are plotted here as mean ± standard deviation between the three replicates, relative to the mean proportion on *Escherichia* OP50 measured on the same day. Significance was calculated using a general linear model with bacteria as a factor and Dunnett's contrasts to compare all conditions against the *Escherichia* OP50 reference (bar highlighted in yellow). Bacterial strains that induce a significant ($P < 0.05$) reduction of *pals-5p*::*GFP* reporter activation are colored in red (darker red for those having a lower than 0.5 relative activation of *pals-5*), those with no significant differences with OP50 in grey, and those with significantly higher activation in blue. Bacteria are arranged according to their phylogenetic relationships, with taxonomic classifications provided in the top rows and a phylogenetic tree based on their 16S sequences in the bottom row. Bacterial strain names in blue indicate a significant phylogenetic signal for these strains. Bacterial strains marked with a grey dot were selected for further investigation.

Our screen included a phylogenetic diverse set of 71 bacterial strains (Fig 1). Using the Local Indicator of Phylogenetic Association (local Moran's I test), we detected no phylogenetic signal for most strains, suggesting that the trait values are randomly distributed across the tree at this level of sampling. However, we identified a significant phylogenetic signal in *Comamonas* strains ($P < 0.01$), indicating a potential phylogenetic clustering for these three strains in

triggering a particularly high *C. elegans* response to viral infection. We also note that all eleven tested members of *Bacteroidota* completely suppressed *pals-5* expression upon OrV exposure.

In four cases, our screen included two distinct bacterial strains labeled with the same species name, indicating a particularly close relationship, and the different strains had similar effects (S4A Fig). We also tested various *E. coli* strains and found that despite some small but significant differences, all enabled a similar proportion of *pals-5p*::*GFP* reporter activation in virus-inoculated animals (S4B Fig).

## In-depth characterization of selected bacteria

From this point onward, we focused on five randomly selected natural bacteria (indicated by grey dots in Fig 1): two enhancing *pals-5p*::*GFP* activation in virus-inoculated animals in the *pals-5* screen and three suppressing it (S2 Fig). We directly stained viral RNA using fluorescence in situ hybridization (FISH) to assess more directly the effect of the bacterial environment on viral infection, and quantified the number of animals harboring virus-infected intestinal cells. On the enhancing *Acinetobacter* BIGb0102 and on the three suppressive bacteria, the viral FISH staining matched the result with the *pals-5* reporter (Fig 2A and 2B). In the *Comamonas* BIGb0172 environment, the proportion of FISH-stained animals remained similar to that in OP50 (Fig 2A).

We further tested on the two enhancing bacteria fluorescent reporters for the *F26F2.1* or *sdz-6* genes that are part of the IPR response but, unlike *pals-5*, are not controlled by the transcription factor ZIP-1 [33]. For *Acinetobacter* BIGb0102 we observed a similar pattern of gene activation for the *F26F2.1* and *sdz-6* reporters as for *pals-5*. However, in the *Comamonas* BIGb0172 environment, the *F26F2.1* reporter but not *sdz-6* was activated at a higher level than on OP50 upon viral infection (Fig 2C).

To assess the levels of viral replication within the animal population, we quantified viral loads using RT-qPCR, collecting the animals 72 h after embryos were placed on the virus-inoculated bacterial lawns. Compared to animals placed on OP50, those on the resistance-inducing bacteria showed a significant and strong ($>$ 250-fold) reduction in RNA1 viral copies. Animals raised on *Comamonas* BIGb0172 showed an 18-fold reduction in viral load compared to those on OP50, and we did not detect significant differences among the viral levels reached on *Escherichia* OP50 and *Acinetobacter* BIGb0102 (Fig 2D).

We tested whether the effect of bacterial environments was specific to the conventional viral strain used in our main experiments (OrV strain JUv1580). We tested JUv2572, an OrV strain reported to be more infectious than JUv1580 and prone to infecting more anterior host intestinal cells [40]. Resistance-inducing bacteria also induced resistance upon inoculation by this viral strain (Fig 2E).

In conclusion, *Chryseobacterium* JUb44, *Sphingobacterium* BIG0116, and *Lelliottia* JUb276 strongly suppressed both viral infection of *C. elegans* and the downstream IPR response. Of the two tested bacteria that enhance the IPR response, *Comamonas* BIGb0172 results in a lower viral load compared to *E. coli* OP50, while the proportion of OrV-infected animals is increased on *Acinetobacter* BIGb0102.

*Caenorhabditis briggsae*, a species related to *C. elegans* and found in similar environments, is also naturally infected by noda-like RNA viruses [27,28]. We tested *C. briggsae* susceptibility to the Santeuil virus on *Chryseobacterium* JUb44 or *Lelliottia* JUb276, which suppress *C. elegans* infection by OrV. Interestingly, these two bacterial environments did not render *C. briggsae* resistant to the Santeuil virus, suggesting that their effect was specific of the *C. elegans*-OrV interaction (Fig 2F).

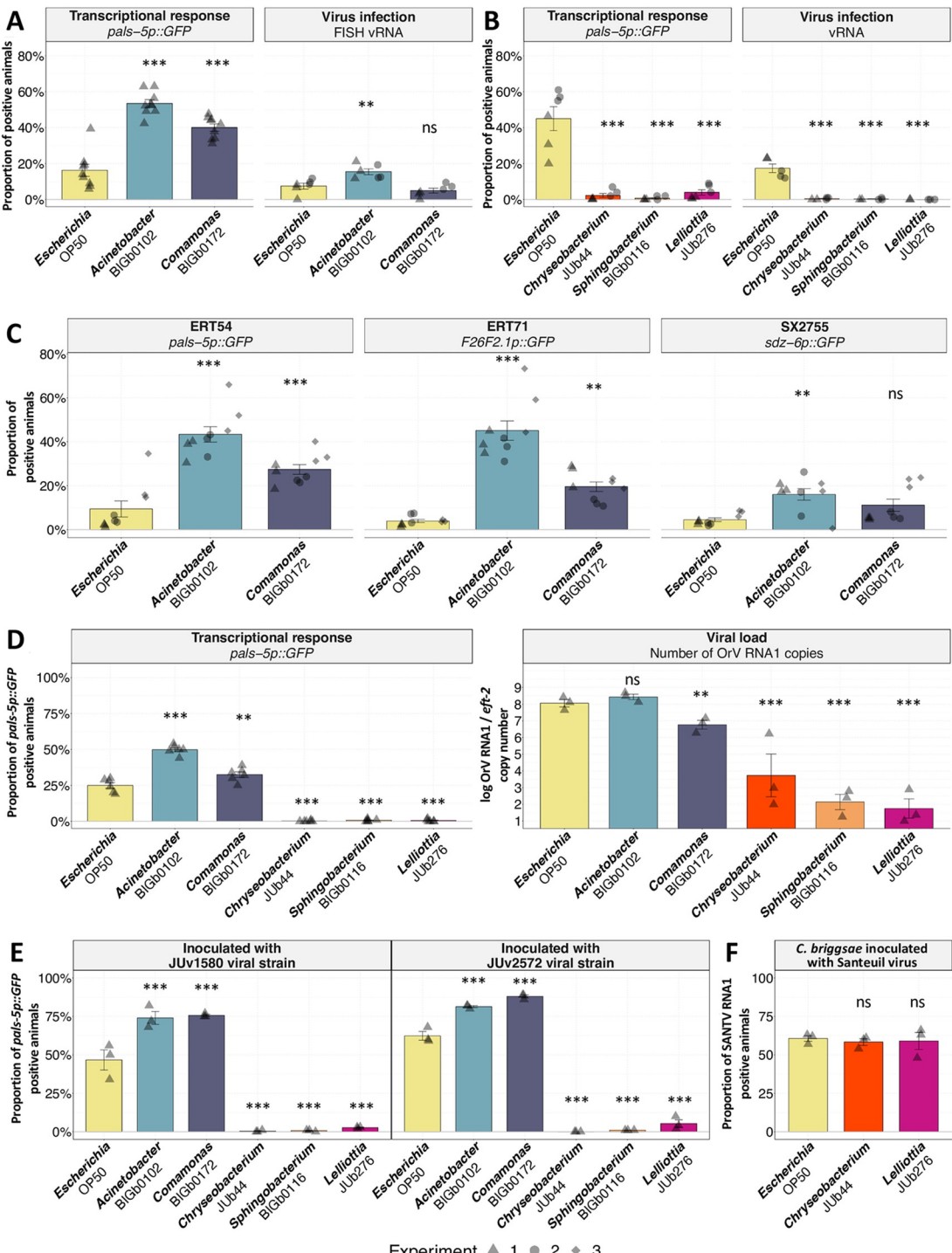

**Fig 2. Focus on five bacterial strains confirms their modulation of *C. elegans* susceptibility to viral infection. (A-B)** Proportion of ERT54 animals showing *pals-5p::GFP* activation (left panel) or infected intestinal cells (stained by FISH against the virus; right panel), after OrV inoculation on bacteria that induced (A) strong activation or (B) no activation at 72 hpi of *pals-5p::GFP* in the initial screen, with *Escherichia* OP50 as a reference. Each bacterial strain is color-coded throughout this and following figures. The proportion of GFP positive animals was assayed on at least 100 animals in 2 independent experiments (different days) with 3 biological replicates per condition each. The two experiments are represented by triangles and circles, respectively. **(C)** Proportion of GFP-positive animals for three transcriptional reporters (*pals-5*, *F26F2.1*, and *sdz-6*), 72 hours post inoculation of Orsay virus in *Acinetobacter* BIGb0102 or *Comamonas* BIGb0172, assayed in three independent experiments, with three replicates of 100 animals each. Each experiment is represented by a distinct shape of the datapoints. **(D)** Viral load at 72 hpi,

measured using RT-qPCR of viral RNA1, in parallel with *pals-5p::GFP* reporter activation measured in six biological replicates of 100 animals each (left panel). From these six populations, three pooled samples were created by combining two populations each. Data normalization was achieved using the copy number of the gene *eft-2* as an endogenous reference (right). **(E)** Proportion of ERT54 animals at 72 hpi, showing *pals-5p::GFP* activation when challenged with different Orsay virus strains on different bacteria, done in three replicates of 100 animals each. **(F)** Proportion of infected *C. briggsae* JU1264 animals at 72 hpi inoculated with Santeuil virus strain JUv1264 on different bacteria, assayed using FISH against the virus. Three biological replicates were evaluated per experiment, with animals per datapoint. Data are presented as mean ± standard error. Black symbols indicate the significance of the difference between the labeled bacteria and the *Escherichia* OP50 reference: *** P < 0.001; ** P < 0.01; * 0.01 < P < 0.05; P values higher than 0.05 are labeled as "ns" (same symbols for all the figures of this study). Significance was calculated for panels 2A-B using a general linear-mixed model where the bacteria was the fixed factor and experiments a random effect; for panel 2C we used a general linear-mixed model with a Gamma distribution and a log-link function in which the bacteria was the fixed factor; Tukey contrasts were used for post hoc analyses. For panels 2D-F we used an analysis of variance with bacteria as a factor and Dunnett's test for post hoc analyses.

## The impact of natural bacteria on *C. elegans* growth rate does not match their effect on viral infection

We wondered whether the bacteria affected OrV propagation by impacting *C. elegans* population and individual growth parameters. We tested the hypothesis of a correlation between *C. elegans* growth parameters on each bacterium and the susceptibility to viral infection.

First, as a proxy of population growth, we measured the production of offspring over time on each of the five bacteria and *E. coli*, in the absence of virus. Animals in the IPR enhancing *Acinetobacter* BIGb0102 had a significant increase in total brood size compared to *Escherichia* OP50 (Fig 3A), while *Comamonas* BIGb0172, *Lelliottia* JUb276, and *Sphingobacterium* BIGb0116 caused a significant decrease. Except for *Acinetobacter* BIGb0102, all bacterial environments significantly delayed the production of offspring compared to *Escherichia* OP50 (Fig 3A), that is, there was no significant difference between the resistant-inducing *Chryseobacterium* JUb44 and the permissive *Comamonas* BIGb0172.

In addition, the natural bacterial environments significantly affected the individual developmental rate (Fig 3B). At 46 h after placing axenic arrested L1 larvae on *Acinetobacter* BIGb0102, the proportion of animals having reached adulthood was higher than those on OP50. The proportion of adults was lower with *Comamonas* BIGb0172, *Lelliottia* JUb276, *Chryseobacterium* JUb44, and *Sphingobacterium* BIGb0116. For the latter two, no adults were observed at this timepoint, while 16 h later in all environments the population was entirely composed of adults.

We thus conclude that the individual and population growth rates on the different bacteria do not match susceptibility to viral infection.

## The effect of resistance-inducing bacteria prevails in mixed bacterial environments

We then wondered whether a suboptimal diet could be the cause of viral resistance. To test this hypothesis, we seeded plates with a mix of individual cultures of a resistance-inducing strain and *Escherichia* OP50. The three tested natural bacteria induced resistance even when initially seeded in a proportion of 10% (Fig 4A).

This allowed us to further test whether the nematode's growth rate mattered for the suppression of viral infection. Following [16], we mixed 20% of the permissive *Acinetobacter* BIGb0102 with 80% of suppressive bacteria that affected *C. elegans* growth rate and observed that developmental rates of the animals were restored (S5 Fig). However, the resistance to viral infection remained (Fig 4B).

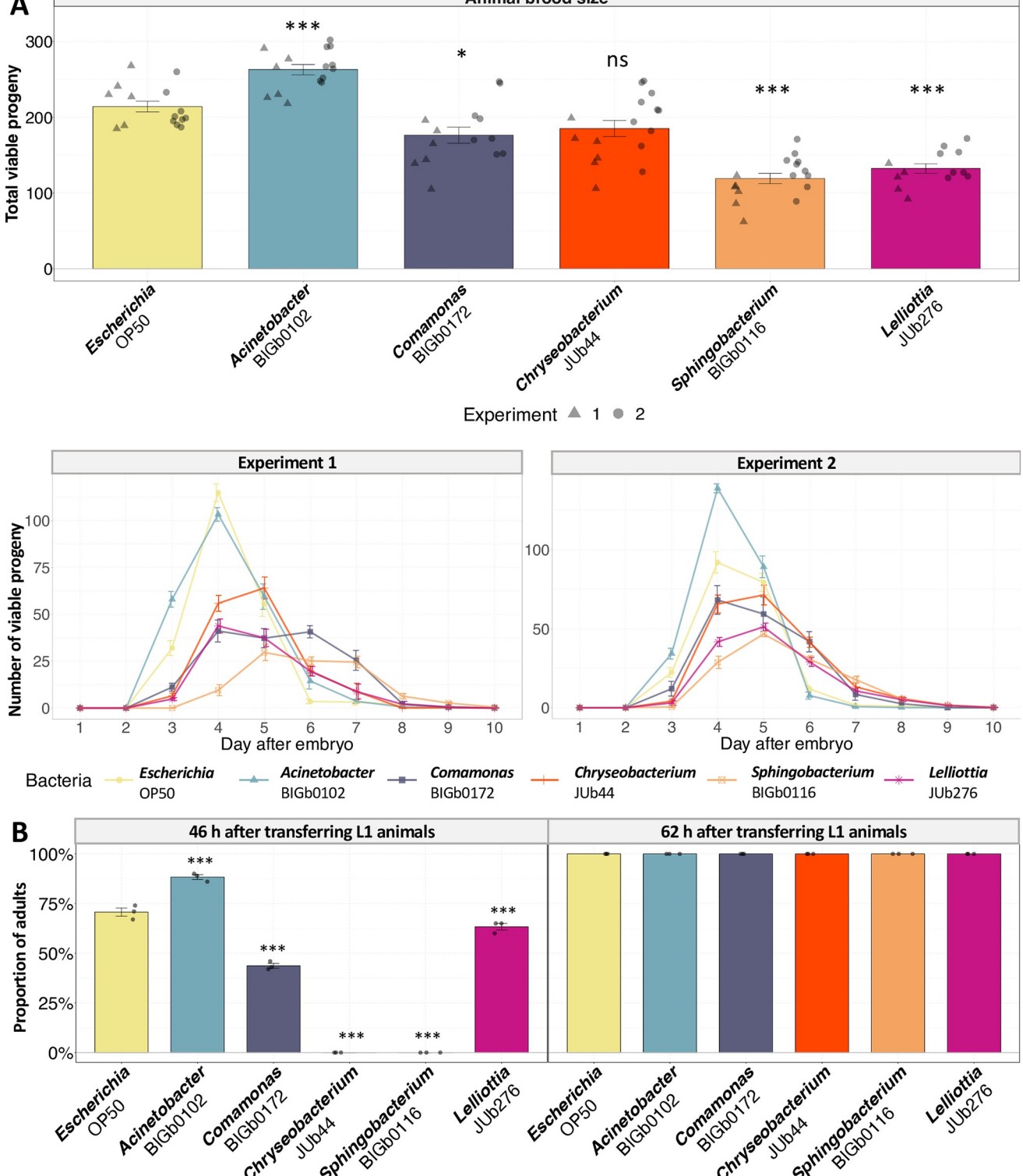

**Fig 3. Impact of selected bacteria on the fitness and developmental rates of *C. elegans* in the absence of viral infection.** (A) Upper panel shows total brood size of non-infected ERT54 animals when placed on each bacterial strain. Lower panels represent the daily production of viable progeny of non-infected ERT54 animals. Two separate experiments were conducted, in which the viable progeny of individual animals was monitored daily, with at least 5 individuals observed per bacterial type in each experiment. The upper panel represents the total progeny of those shown in the lower panels; the two experiments are represented by triangles and circles, respectively. (B) Proportion of adults assayed after exposing arrested axenic L1 larvae of the ERT54

strain to each bacterium, after 46 h (left panel) and after 62 h (right panel), in a population of 100 animals in each of three replicate populations per bacterial strain. Data are presented as mean ± standard error. Black symbols on the graphs indicate the statistical significance of differences when compared to the *Escherichia* OP50 reference: *** P < 0.001; ** P < 0.01; * 0.01 < P < 0.05. P-values exceeding 0.05 are labeled as "ns". Significance was calculated for panel 3A using a linear-mixed model where the bacteria was the fixed factor with experiment as random effect; Tukey contrasts were used for post hoc analyses. For panels 3B we used an analysis of variance with bacteria as a factor and Dunnett's test for post hoc analyses.

Seeding plates with a mix of bacteria could cause potential interactions and competition between them. To minimize bacterial interaction, we (i) prepared plates with the suppressive natural bacteria and added permissive bacteria: *Escherichia* OP50 (Fig 4C) or *Acinetobacter* BIGb0102 and *Comamonas* BIGb0172 (Fig 4D), right before transferring axenic embryos and inoculating OrV or (ii) added suppressive natural bacteria onto *Escherichia* OP50 plates (Fig 4E and 4F). On these supplemented environments, *Chryseobacterium* JUb44, *Sphingobacterium* BIGb0116, and *Lelliottia* JUb276 induced resistance to infection. However, the induced OrV resistance was abolished if the suppressive natural bacteria were first heat-killed (Fig 4E) or when the filtered supernatant of live bacteria was added (Fig 4F).

The prevalence of resistance-inducing effects was also observed when testing the CeMbio community, a defined natural and ecologically relevant bacterial community [41]. Individual bacterial strains in the CeMbio community had diverse effects on susceptibility to infection. However, the animals were resistant to infection when associated with the CeMbio community (S6A Fig). Thus, bacterial-induced resistance prevails over permissive bacteria. These findings thus reject the poor diet hypothesis.

## Suppression of viral infection is not explained by avoidance of the bacterial lawn

*C. elegans* avoids certain bacteria [42]. In our experiments, we applied the viral inoculum atop a bacterial lawn spotted in the center of the agar plate. We hypothesized that the animals may avoid this lawn, hindering virion uptake. To investigate this hypothesis, we analyzed the distribution of animals on plates spotted with a central bacterial lawn. The animals displayed no avoidance during the initial 48 hours. Beyond this period, the animals exhibited some aversion to the resistance-inducing *Lelliottia* JUb276 and to the susceptibility-inducing *Acinetobacter* BIGb0102. This behavior was observed in both mock and virus-inoculated plates (Fig 5A). These results indicate that the level of aversion to the bacteria did not match their effect on viral infection.

To further ensure that the avoidance of the viral inoculum was not responsible for the absence of infection on suppressive bacteria, we distributed both virus and bacteria on the whole surface of the agar. In this condition where the animals could not avoid exposure to the bacteria and the virus inoculum, bacterial strains JUb44, BIGb0116, and JUb276 still conferred viral resistance (Fig 5B).

## Natural bacteria eliminate OrV from pre-infected *C. elegans* populations within two generations

In natural environments, viral infections spread within populations as infected organisms continuously produce and release viruses [43]. In order to create conditions where horizontal virus transmission must occur for the viral infection to be maintained, we transferred individuals from an infected *C. elegans* population, previously inoculated on *Escherichia* OP50, to the resistance-inducing bacterial environments (Fig 6A). The offspring of these animals had reduced activation of the *pals-5p::GFP* infection reporter and a lower proportion of infected

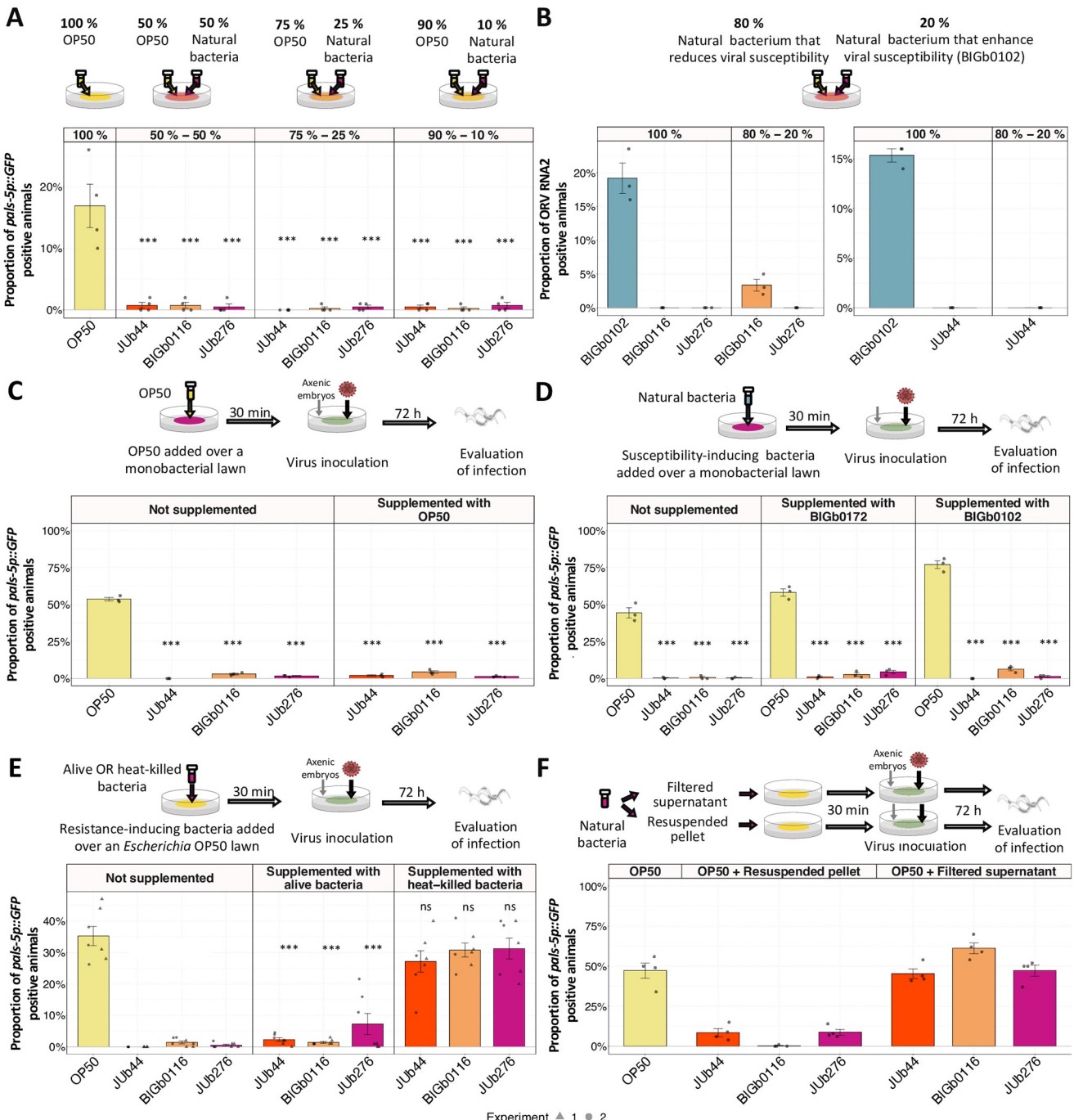

**Fig 4. Prevalence in mixed bacterial environments of the suppressive effect of viral infection.** In all panels the infection was evaluated at 72 hpi. **(A-F)** Activation of *pals-5p::GFP* reporter or FISH staining of viral RNA2 (panel B) after OrV inoculation of ERT54: **(A)** on bacterial lawns seeded with dual combinations of resistance-inducing bacteria and OP50 in the indicated proportions (assayed in four replicates of 100 animals each); **(B)** on bacterial lawns seeded with a 80–20% combination of resistance-inducing bacteria and BIGb102, a bacterial strain permissive for viral infection (assayed in three replicates of 100 animals each); **(C)** on a lawn of resistance-inducing bacteria supplemented with OP50 (assayed in three replicates of 100 animals each); **(D)** on a lawn of resistance-inducing bacteria supplemented with BIGb102; **(E)** on a lawn of *Escherichia* OP50 supplemented with live or heat-killed cultures of resistance-inducing natural bacteria (assayed in two independent experiments, one with three replicates and the other with four replicates of 100 animals each—each experiment is represented by a distinct shape of the datapoints); **(F)** on *Escherichia* OP50 supplemented with resuspended pellet or filtered supernatant of resistance-inducing natural bacteria (assayed in four replicates of 100 animals each). In all panels, the top row shows a schematic representation of the experimental design. Data are presented as mean ± standard error. Black symbols indicate the significance of the difference between the labeled bacteria and the *Escherichia* OP50 reference: *** P < 0.001; ** P < 0.01; * 0.01 < P < 0.05; P values higher than 0.05 are labeled as "ns" (same symbols for all the figures of this study). In panels A-D and panel F significance was calculated using a general linear model where the factors were bacteria

and treatment. In panel E significance was calculated using a general linear-mixed model, where the factors were the bacteria and the treatment, and the experiment was considered a random effect. In both cases Tukey contrasts were used for post hoc analyses.

individuals, compared to the control *Escherichia* OP50. At the second generation, viral infection was almost eliminated, except for the control (Figs 6B and S6B). Another experiment where the transfer was performed by transferring a piece of agar from the first plate to a new plate gave a similar result (S6C Fig).

## Resistance-inducing bacteria act after the ingestion of virions

To examine whether the bacterial lawn affected virion infectivity prior to entering the host, we employed two approaches: (i) pre-incubating the virus with the bacteria before inoculation, and (ii) inoculating *C. elegans* with the virus on an *Escherichia* OP50 lawn and then transferring them to another bacterial environment.

In the first approach, we incubated the virus with either *E. coli* or the suppressive bacteria at 20°C for 24 h (Fig 7A). After incubation, bacteria were pelleted and the filtered viral supernatants were inoculated to animals on *Escherichia* OP50. We did not see a significant difference in the final infectivity of the viral preparations (Fig 7A). These results suggest that the bacteria do not induce resistance to infection by degrading the virions outside the host.

In the second approach, we exposed axenic embryos to OrV for 24 h on *Escherichia* OP50 lawns and then transferred the larvae to plates with different bacterial lawns (Fig 7B). Note that in this experimental setup the animals acquire the virus in the same bacterial environment, thus pumping the virus at the same rate. We observed that natural bacteria can alter the *pals-5p*::*GFP* activation and infection susceptibility (proportion of infected animals), after initial virus exposure in a common susceptible environment (Fig 7B). These results indicate that the tested bacteria induce resistance after the ingestion of the virus into the intestinal lumen of the animal and that pumping rate is not a critical factor for the observed resistance.

## Intact host antiviral pathways are not required for bacterial suppression of viral infection

We sought to identify host pathways required for the suppressive effect of bacteria on viral infection, testing animals carrying null alleles for key components of the antiviral response. A main axis of antiviral immunity starts with DRH-1/RIG-I triggering both the small RNA response and the transcriptional IPR (therefore lowering the ubiquitin-mediated immunity). As the IPR response is not activated in *drh-1* mutants, we could not use *pals-5p*::*GFP* as the reporter and instead utilized a *lys-3p*::*GFP* reporter, which is activated upon severe infection [34]. Upon OrV inoculation, 75% of *drh-1* animals activated the *lys-3p*::*GFP* reporter on *Escherichia* OP50 but very few to none did so on the 28 bacteria tested (Fig 8A). Thus, the suppression is mostly independent of DRH-1, a main node in the antiviral response.

On three bacteria (*Raoultella* BIGb0399, *Pseudomonas* BIGb0477, and *Acinetobacter* MYb10), the activation of *lys-3p*::*GFP* was less strongly reduced (Fig 8A). We repeated the experiment using viral RNA FISH on two of the latter bacteria (Fig 8B and 8C) and the three suppressive bacteria studied above (Fig 8D and 8E).

On the *Chryseobacterium* JUb44 and *Sphingobacterium* BIGb0116, the virus could not infect the *drh-1* mutant (Fig 6D), indicating that these bacteria do not require DRH-1 to repress OrV infection.

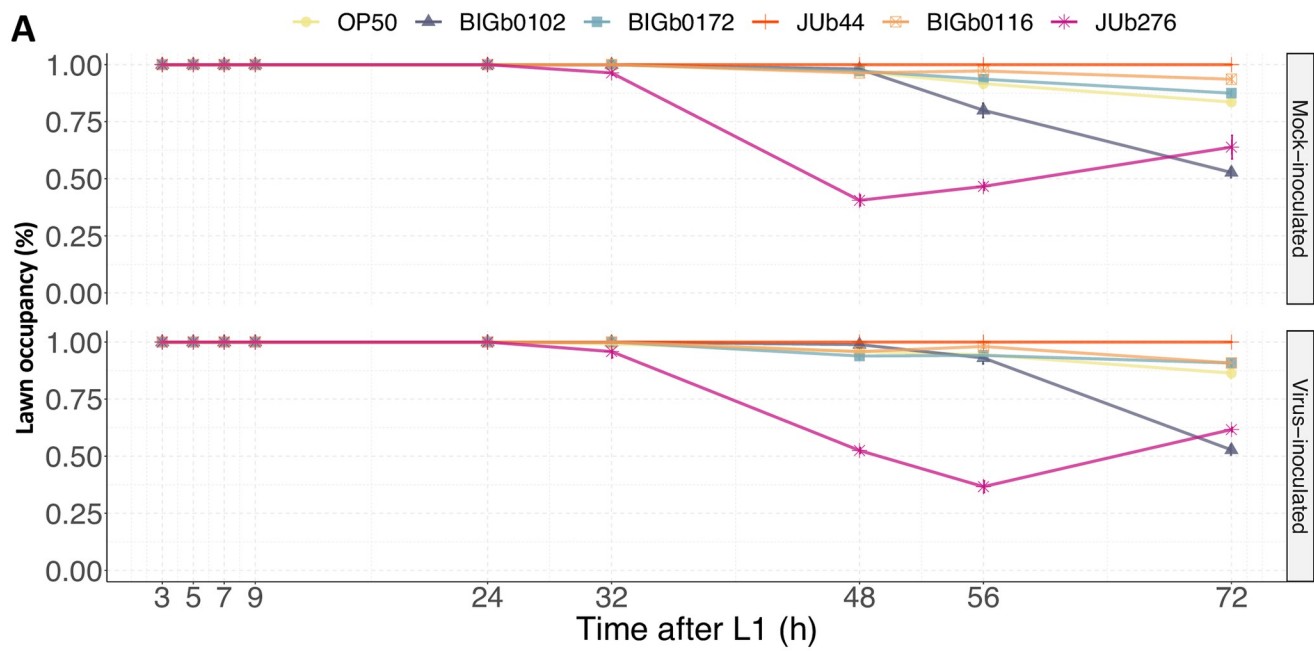

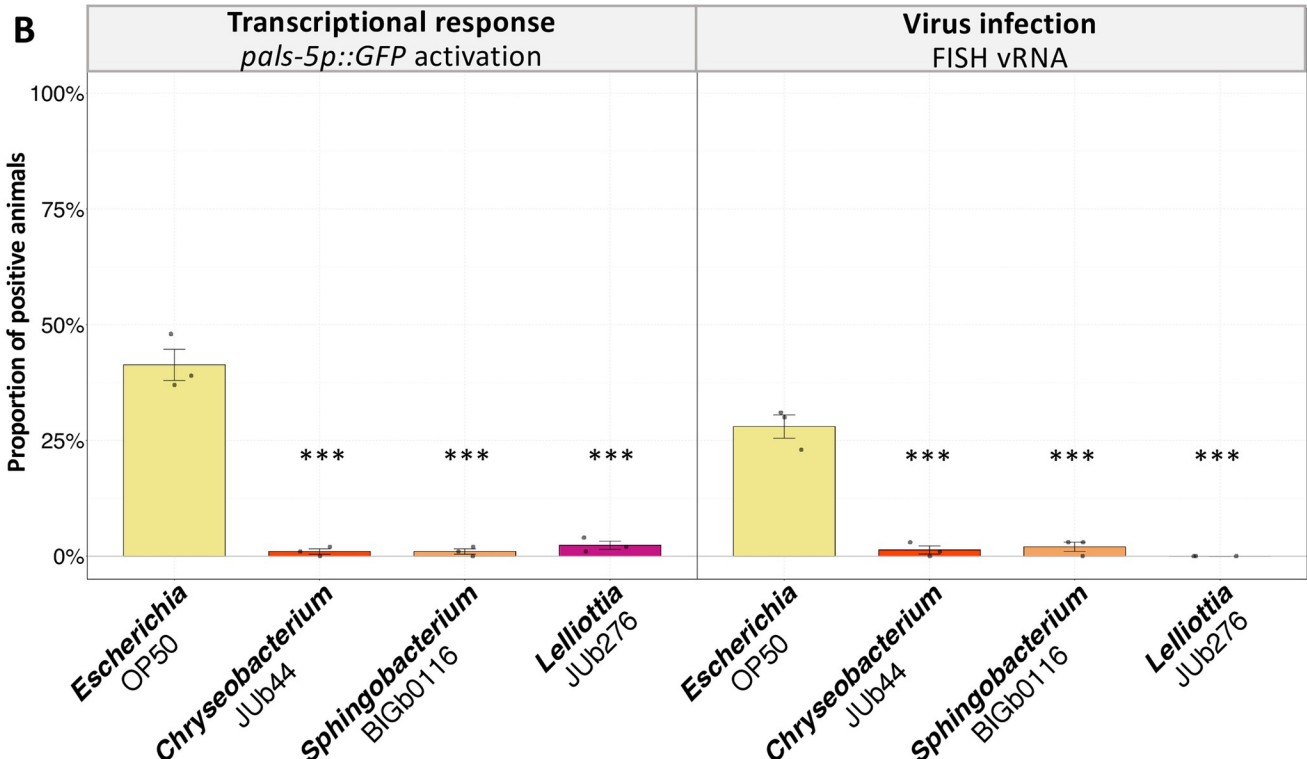

**Fig 5. Aversion to the bacterial lawn does not explain the suppression of viral infection. (A)** Around 100 axenic arrested L1 larvae of the ERT54 strain were placed around different bacterial lawns that where mock-inoculated with M9 (upper panel) or inoculated with OrV JUv1580 (lower panel). The proportion of individuals occupying the bacterial lawn was visually counted at different timepoints. **(B)** Proportion of animals showing a transcriptional IPR response (*pals-5p*::*GFP* activation; left panel) or infected intestinal cells (stained by FISH against the virus; right panel) at 72 hpi in plates fully covered by bacteria and the JUv1580 virus inoculum. Three biological replicates were tested per bacteria and 100 animals from each were assayed. Data are presented as mean ± standard error. Black symbols indicate the significance of the difference between the labeled bacteria and the *Escherichia* OP50 reference: *** P < 0.001; ** P < 0.01; * 0.01 < P < 0.05. P-values greater than 0.05 are labeled as "ns". Significance was calculated using an analysis of variance with bacteria as a factor and Dunnett's test for post hoc analyses.

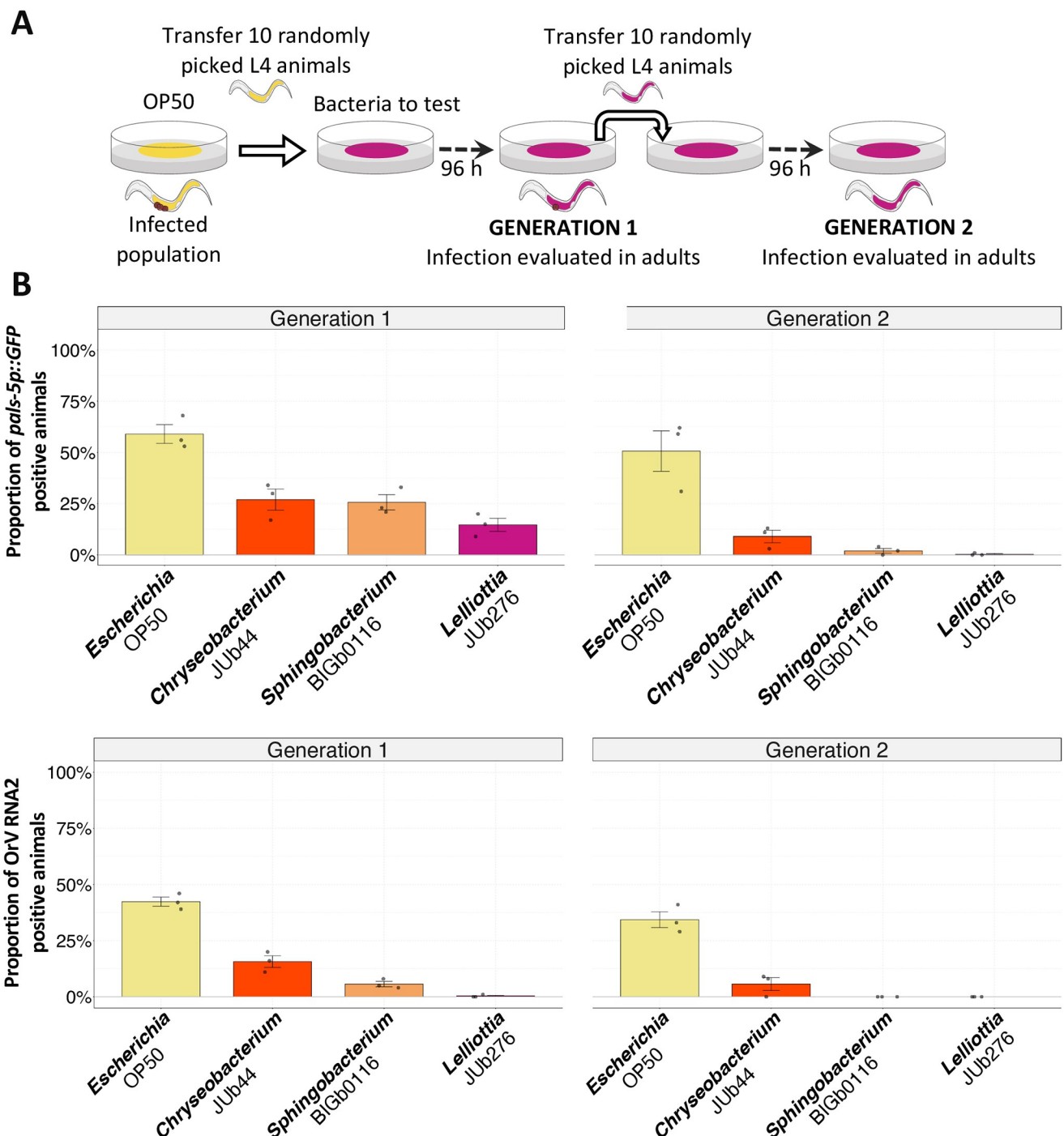

**Fig 6. Bacterial environments can suppress OrV persistence over generations. (A)** Schematic representation of the experimental design (detailed in methods). ERT54 animals carrying the *pals-5p*::*GFP* reporter were inoculated with OrV JUv1580 on *E. coli* OP50 and transferred to selected bacteria. **(B)** Activation of the *pals-5p*::*GFP* reporter (upper panel) or proportion of infected animals (stained using FISH; lower panel) at two successive generations. Datapoints represent 100 animals, with three biological replicates per bacterial strain. The bar represents the mean ± standard error among replicates. Black symbols indicate the significance of the difference between the labeled bacteria and the *Escherichia* OP50 reference: *** P < 0.001; ** P < 0.01; * 0.01 < P < 0.05. P-values greater than 0.05 are labeled as "ns". Significance was calculated using a general linear model where bacteria and generation were the factors. Tukey contrasts were used for post hoc analyses.

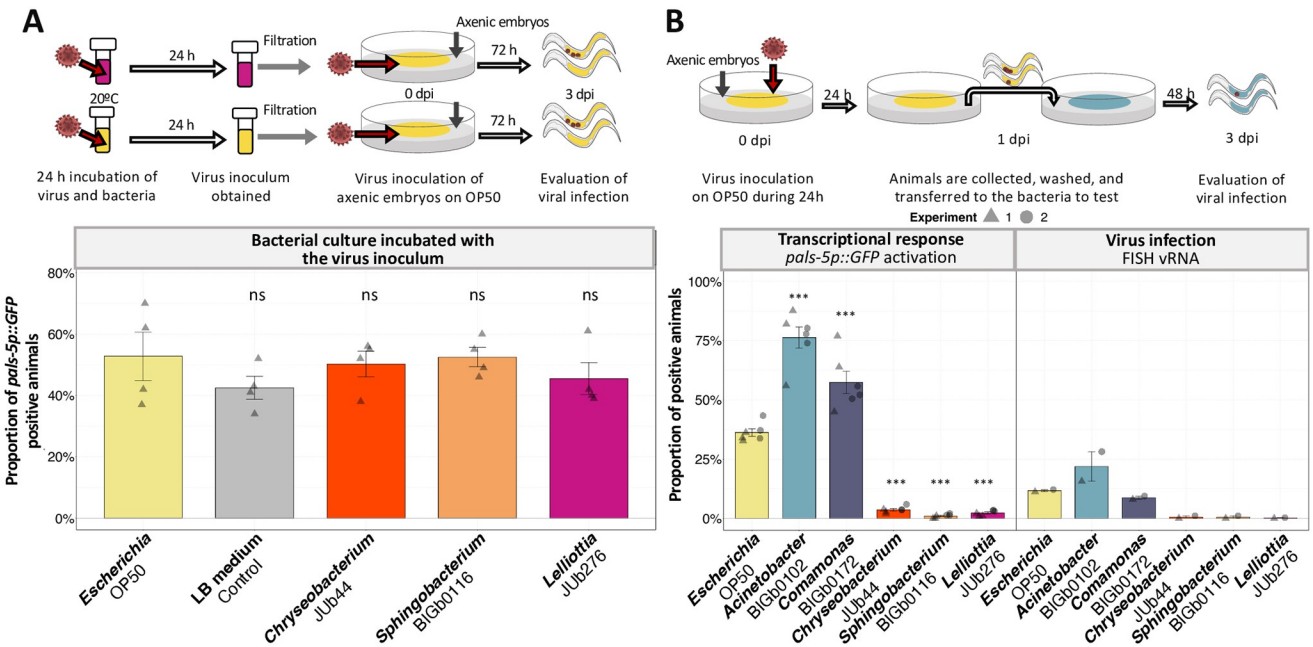

**Fig 7. Suppressive bacterial environments act after ingestion of virus. (A)** Activation of the *pals-5p*::*GFP* reporter in ERT54 animals on *Escherichia* OP50, challenged with viruses previously incubated with various bacterial cultures (detailed in Methods section "Coincubation of bacterial culture and virus inoculum"). Four replicate populations were evaluated per condition, with at least 100 animals assayed per population. Upper panel shows a schematic representation of the experimental design (detailed in Methods). **(B)** Proportion of GFP-positive ERT54 animals after initial exposure to the virus on *Escherichia* OP50 and subsequent transfer to different, non-virus-inoculated, bacteria. The upper panel shows a schematic representation of the experimental design (detailed in Methods section "Common garden inoculation experiment"). For the transcriptional response, three replicate populations were evaluated per condition and experiment. These three populations were pooled and the vRNA FISH stained to quantify the proportion of infected animals. Each data point represents a biological replicate, with at least 100 animals assayed per population. Independent experiments performed on different days are represented by the shape of the data point. Data are presented as mean ± standard error. "dpi" = days post-inoculation. Black symbols indicate the significance of the difference between the labeled bacteria and the *Escherichia* OP50 reference: *** P < 0.001; ** P < 0.01; * 0.01 < P < 0.05; P values higher than 0.05 are labeled as "ns". Significance was calculated for panel A using an analysis of variance with bacteria as a factor and Dunnett's test for post hoc analyses and for panel B using a general linear-mixed model where bacteria was the fixed factor with experiments as random effect and Tukey contrasts for post hoc analyses.

On the *Pseudomonas* BIGb0477, *Acinetobacter* MYb10, and *Lelliottia* JUb276 environments, the *drh-1* animals showed a lower infection than on *Escherichia* OP50, but a significantly increased infection compared to the wild-type animals (Figs 8B, 8D, 8G and S6D). We confirmed the results for *Lelliottia* JUb276 using a *drh-1* allele mimicking a natural deletion [31] (S6E Fig). Thus, even for these three bacteria, a suppressive effect is observed in the absence of *drh-1*, but the suppression is incomplete.

Downstream of DRH-1, we tested mutants in RDE-1, the Argonaute required for the RNAi interference response, and in the ZIP-1 transcription factor required for part of the DRH-1 dependent transcriptional IPR (Fig 8C, 8E and 8F). On these mutants, the suppressive bacteria, including *Lelliottia* JUb276, fully suppressed infection. Thus, the weak infection observed in *drh-1* mutants on *Lelliottia* JUb276 does not seem to be RDE-1 or ZIP-1 dependent.

We further tested *cde-1* mutants, which are defective in the viral uridylation response, and found that *cde-1* mutants are resistant to viral infection on *Chryseobacterium* JUb44, *Sphingobacterium* BIGb0166, and *Lelliottia* JUb276 environments (Fig 8F).

Finally, we wondered if immune pathways against bacteria, such as p38/PMK-1, may be involved in the viral resistance induced by bacteria. We tested the susceptibility to viral infection of *tir-1*, *tol-1*, and *pmk-1* mutants on *Lelliottia* JUb276. This bacterium induced resistance

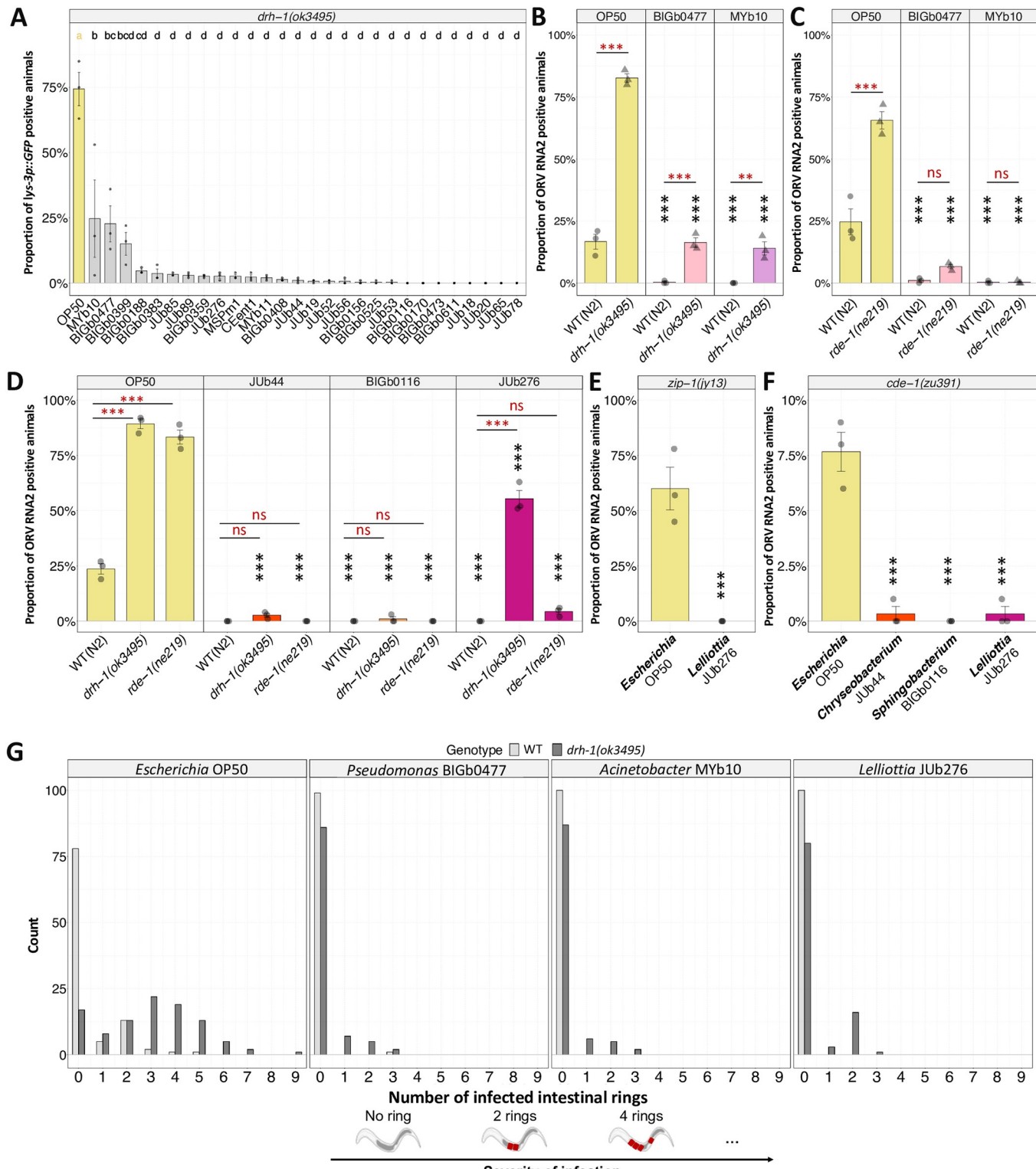

**Fig 8. Bacterial environments also suppress OrV infection in animals with hampered antiviral immune responses. (A)** Proportion of *drh-1* animals that activate the *lys-3p::GFP* reporter after OrV inoculation on different viral resistance-inducing bacteria. **(B-C)** Proportion of infected animals, as assayed by RNA2 FISH, on two bacteria that enable slight activation of the reporter in panel A, for (B) wild-type and *drh-1*; (B) wild-type and *rde-1* animals. **(D-F)** Proportion of infected animals, as assayed by RNA2 FISH, on the three selected suppressive bacteria for (D) wild-type, *drh-1* and *rde-1*; (E) *zip-1*; (F) *cde-1* animals. **(G)** Number of OrV-infected intestinal rings, as assayed by RNA2 FISH, in WT and *drh-1* animals on the. In panels (A-F), three replicate populations were

evaluated per condition. Each data point represents a replicate of at least 100 animals. Data are presented as mean ± standard error. In panel G three populations were pooled and 100 animals were evaluated. Black symbols indicate the significance of the difference between the labeled bacteria and the *Escherichia* OP50 reference meanwhile red symbols indicate the significance of differences between genotypes: *** $P < 0.001$; ** $P < 0.01$; * $0.01 < P < 0.05$; P values higher than 0.05 are labeled as "ns". Significance was calculated using an analysis of variance with bacteria as a factor (Fig 8A, 8E, and 8F) or a general linear model where the factors were bacteria and host genotype (Fig 8B, 8C, and 8D). In both cases Tukey contrasts were used for post hoc analyses.

in all the mutants we tested (S7A and S7B Fig). In addition, we tested if the natural bacteria we studied in depth would activate a *sysm-1* reporter, an element of the p38/PMK-1 pathway. None of the tested bacteria activated the reporter (S7C Fig).

In conclusion, for most suppressive bacteria, the reduction in viral infection is independent of known antiviral responses including small-RNA mediated RNA degradation and transcriptional regulation downstream of DRH-1, and uridylation. However, *Pseudomonas* BIGb0477, *Acinetobacter* MYb10, and *Lelliottia* JUb276 fail to induce full resistance to viral infection when DRH-1 is not functional.

## Discussion

We conducted a comprehensive survey of the impact of 71 bacteria naturally associated bacteria with *C. elegans* on its susceptibility to viral infections. Our screen revealed that monobacterial environments significantly influence host susceptibility to viral infections, with some bacteria providing protective effects, while others are permissive of viral infections. Notably, in our screen, the majority of the tested natural bacteria tested reduced host susceptibility to viral infection compared to the *E. coli* OP50 environment commonly used in laboratories. However, when we tested two of these suppressive bacteria on the infection of *C. briggsae* by its Santeuil virus, they did not induce resistance, suggesting specificity in interaction either with the host, the virus or both. Overall, our results highlight the importance of considering the laboratory microbial environment when isolating wild animal strains for the study of their viruses.

Host-microbe interactions include possible trade-offs for the host between the induced viral resistance and the effect of the bacteria on the host, regardless of virus presence [44]. We found that the BIGb0116 and JUb276 bacterial strains may confer protection against viral infections while simultaneously reducing exponential growth parameters of host populations in the absence of virus. In the five natural bacterial environments we tested in depth, except for BIGb0102, animals spend more time in their larval stage due to delayed development and the populations thus grow more slowly. The developmental delay does not seem to explain the resistance to infection, as all four *C. elegans* larval stages are susceptible to infection, as has been shown for the JU1580 [45] and N2 [46] strain. Most convincingly we observed that the resistance to viral infections is maintained in mixed bacterial environments that restore the animal development rates.

Host nutrition can have diverse effects in pathogens [47]. We have grown *C. elegans* in different bacterial environments and it is important to remark that for this nematode there is no distinction between food (nutrition) and biotic environment. Our experiments suggest that the resistance to virus infection is not caused by nutritional deficits because supplementation with bacteria that enable virus infection does not recover susceptibility to the virus. Supplementation of *C. elegans* with chemicals has been shown to restore viral susceptibility on resistant genotypes; mutants with disturbed lipid metabolism restore susceptibility to infection after being supplemented with certain lipids [48]. Some bacterial strains have been shown to possess immunomodulatory properties even after being heat-killed [49]. Our tested bacteria do not induce resistance after being heat-killed, indicating that either the bacteria need to be alive to induce resistance or that the bacterial factor that induces resistance is thermolabile.

We conclude that the influence of natural bacteria on *C. elegans* susceptibility to virus is beyond nutritional content. This observation aligns with observations made for other physiological phenotypes of the nematode [16].

We found that some bacteria can induce resistance to infection without requiring known host antiviral mechanisms. Some bacteria fail to induce full resistance in *drh-1* null mutants but could induce it in mutants of factors acting downstream of DRH-1. DRH-1 recognizes the viral genome as foreign, possibly through its well-conserved RIG-I domain [31], and activates multiple immune responses and transcription of genes of unknown function. Our findings suggest that either some bacteria fail to induce resistance in the highly susceptible *drh-1* mutants or that the bacteria instigate resistance partially through DRH-1, via an unknown downstream pathway. It is also possible that rather than through DRH-1 mediated activation of antiviral pathways, the suppression operates because viral RNA does not enter the intestinal cells on suppressive bacteria, or that the cells are not competent for some part of the lifecycle of the virus, such as replication, translation or packaging.

Our research emphasizes the importance of considering the natural biotic environment in the study of viral infections and their ecological and evolutionary implications [50,51]. Our work is consistent with a recent study by Vassallo and colleagues that reported that *Pseudomonas lurida* and *Pseudomonas aeruginosa* attenuate Orsay virus transmission and infection rates. This attenuation depends on bacterial regulators of quorum sensing [52]. Our study provides a valuable foundation for future research by suggesting the involvement of unknown antiviral mechanisms and by establishing a basis for dissecting the molecular mechanisms underlying host-virus interactions accounting for the biotic environment. Future research should aim to uncover the molecular mechanisms underlying these protective effects and explore their applicability to a broader range of viral strains and host species. This understanding may aid in discovering new immune mechanisms and improving host health through targeted manipulation of the microbes associated with an organism.

## Material and methods

### Animal strains and maintenance

*C. elegans* was cultured on Nematode Growth Media (NGM) plates at 20˚C, following standard procedures [53,54]. *C. briggsae* was cultured at 23˚C. The NGM plates were seeded with 100 μL of a monobacterial culture and kept at room temperature for two days before being stored at 4˚C. All experiments were performed using seeded plates stored for less than 3 weeks. The nematode strains used in this study are listed in S2 Table.

The *drh-1(mcp553)* allele mimics the natural deletion in JU1580 and other *C. elegans* wild isolates [31]. It was created using a CRISPR/Cas9-mediated edition in the N2 background by the CNRS Segicel Platform (Lyon, France) using two guides crMG023 (gCTATCGTGTTGC TAGTCGA)and crMG024 (ACCGACCGAAATACGACAAT) and the repair template (tctttacatgcttattttatttaattcttaattctattaattatttaattttcagctatc AATGAGAGATGCGGAT CAAGCTCGAACACCAATGGTATTTGAGCATCACGCGAATGGAGA). The primers used to confirm the deletions were AAACTCGCCTGACGGATGAG, TTGGAACTGAGC GATTGGCA, and TCGGTACCTTCGACTAGCAAC.

The JU4289 strain with the *agIs219[sysm-1p::GFP + ttx-3p::GFP]* transgene in the N2 background was obtained by crossing ZD39 hermaphrodites *[agIs219 [sysm-1p::GFP + ttx-3p::GFP] III; pmk-1(km25) IV]* to N2 males, then F1 males to N2 and selecting a GFP-positive animal.

## Bacterial maintenance

Bacteria were cryo-preserved at -80˚C in Luria broth medium (LB; 10 g/L tryptone, 5 g/L yeast extract, 5 g/L NaCl) with 25% glycerol. For cultivation, a cryo-preserved culture was streaked on LB-agar plates and incubated at room temperature for 48 h. Colonies were then picked and inoculated into 5 mL of liquid LB. The culture was grown at 28˚C and 220 rpm for 16 h for the naturally-associated bacterial strains, while *E. coli* OP50 was grown at 37˚C. The bacterial strains used in this study are listed in S1 Table.

## Preparation of viral inoculum

The Orsay virus strain JUv1580 [27] was used in all experiments except when the Orsay virus strain JUv2572 [40] or the Santeuil virus JUv1264 [27] were used, as indicated. Viral preparations of JUv1580 were obtained by inoculating a viral filtrate derived from the original JU1580 infected animals [27] on 90 mm OP50-seeded plates with JU2624 animals (*C. elegans* JU1580 isolate in which the *lys-3p*::*GFP* construct was introgressed by 10 rounds of backcrosses to JU1580; 34). The animals were collected with M9 buffer four days after inoculation, pelleted, and the supernatant was filtered through a 0.22 μm filter to obtain the OrV-infectious supernatant. The supernatant was aliquoted and cryo-preserved at -80˚C and the required amount was freshly thawed at each experiment.

## Virus inoculation procedure

To synchronize the nematode population and obtain axenic animals, we treated young adult populations with 4 mL of a bleach solution (2 mL sodium hypochlorite 12%, 1.25 mL NaOH 10 N, 6.75 mL H2O) for three minutes. We then washed the samples four times with 15 mL of M9 buffer. This procedure resulted in axenic embryos, which were placed around the bacterial lawn previously inoculated with 50 μL of the viral inoculum. The inoculated plates with animals were maintained at 20˚C, and infection was assessed 72 hours post-inoculation (hpi).

## Fluorescent reporters for viral infection

Animals were observed using a Leica MZ FLIII fluorescence stereomicroscope at 6x magnification. An animal was visually classified as positive when GFP fluorescence was observed in intestinal cells at higher levels than background. The GFP signal was scored as a binary trait, quantified over the population of animals. The activation of the intracellular pathogen response was measured using fluorescent reporters activated upon intracellular infection (*pals-5p*::*GFP*, *F26F2.1p*::*GFP*, or *sdz-6p*::*GFP*; 33,34,39). In the genetic background *drh-1*, a reporter of severe biotic stress (*lys-3p*::*GFP*) was used, as *drh-1* animals do not induce *pals-5* expression.

## Fluorescent *in situ* hybridization

The proportion of infected animals was quantified by visualizing viral RNA in the host intestinal cells. Viral RNA was labeled using fluorescent in situ hybridization (FISH), as described by Frézal and colleagues [40]. In brief, animals were collected and fixed in a 10% formamide solution. Fixed animals were stained targeting OrV RNA2 molecules using a 1:40 dilution of a mix of oligonucleotide sequences [40,55] conjugated to the Cal Fluor red 610 fluorophore or a non-diluted single probe (5' ACC ATG CGA GCA TTC TGA ACG TCA 3') conjugated to Texas Red. To stain the Santeuil virus we targeted its RNA1 molecules using a 1:40 dilution of a mix of oligonucleotide sequences [40] conjugated to the Cal Fluor red 610 fluorophore. Animals were then examined using an AxioImager M1 (Zeiss) compound microscope with 10×

(0.3 numerical aperture) and 40× (1.3 numerical aperture) objectives. An animal was considered infected if at least one intestinal cell displayed distinct fluorescence at higher levels than the background staining for this individual.

## RNA extraction

RNA was isolated following an acid guanidinium thiocyanate-phenol-chloroform extraction protocol [56]. A 100 µl *C. elegans* pellet was resuspended in 500 µl of Trizol, and the suspension was subjected to 5 freeze-thaw cycles. The suspension was then vortexed for 30 seconds, allowed to rest for an equal duration, and this vortex-rest sequence was performed five times. 100 µl of chloroform were then added. Tubes were shaken vigorously by hand for 15 seconds and left to stand at room temperature for 2–3 minutes. Following incubation, the mix was centrifuged for 15 minutes at 13000 rpm at a temperature of 4°C. The upper aqueous phase was transferred to a new tube, where 250 µl of isopropanol was added, mixed, and allowed to incubate at room temperature for 10 minutes. A subsequent centrifugation was performed under the same conditions. The resulting supernatant was discarded, and the pellet was washed using 500 µl of 75% ethanol. This mix was then vortexed and centrifuged for 5 minutes at 13,000 rpm at 4°C. After discarding the supernatant, the pellet was air-dried for 10 minutes and dissolved in 50 µl of RNAse-free water.

## Measurement of viral loads

cDNA was generated from 500 ng of total RNA with random primers using cDNA that was synthesized with SuperScript IV Reverse Transcriptase (Thermo Fisher Scientific, Waltham, MA, USA), following manufacturer's instructions. cDNA was diluted to 1:10 for RT-qPCR analysis. RT-qPCR was performed using LightCycler 480 SYBR Green I Master (Roche, Mannheim, Germany), following manufacturer's instructions. The amplification was performed on a LightCycler 480 Real Time PCR System (Roche). In each sample the viral RNA1 (primers GW194: 5′ ACCTCACAACTGCCATCTACA and GW195: 5′ GACGCTTCCAAGATTGG TATTGGT, 27) levels were measured and normalized to its corresponding levels of the endogenous gene *eft-2* (etf-2 2F 5′ CTGCCCGTCGTGTGTTCTAC and etf-2 2R 5′ TCCTCGAAAACGTGTCCTCTT).

## Mixed bacterial environment experiments

The following two approaches were taken to mix bacteria. (i) For Fig 4A and 4B: volumes of liquid bacterial cultures of suppressive and permissive strains were mixed according to the proportions indicated in the figures, and 100 µL of the mixture were seeded on NGM plates. These plates were maintained at room temperature for two days before being stored at 4°C until use. Plates were utilized within three weeks and acclimated at room temperature for a few hours before the start of the experiment. (ii) For Fig 4C to 4F: 30 minutes before initiating the experiment, we added 100 µL of bacterial culture (prepared as described in "Bacterial maintenance") atop the bacterial lawn of plates seeded with a monobacterial lawn (prepared as described in "Animal strains and maintenance"). Specifically, in Fig 4E a heat-killed culture was added to the seeded plates. Bacteria were heat-killed by incubating the bacterial cultures in a water-bath at 100°C for 40 min. The effectiveness of the heat-killing process was confirmed by plating 100 µL of the heat-killed culture onto an LB plate and observing no growth after overnight incubation at 37°C. In Fig 4F, we pelleted 5 mL of a liquid bacterial culture by centrifuging it for 10 minutes at 5000 rpm. The supernatant was collected and filtered through a 0.22 µm filter. Subsequently, 100 µL of the filtered supernatant was added atop a

monobacterial lawn. The pellet was resuspended in 5 mL of LB and 100 µL of the resuspended pellet was added on the top of a monobacterial lawn, as indicated in the *x* axis of the figure).

### Experiments assaying OrV persistence over generations

We initiated these experiments with ERT54 animal populations infected with OrV, as indicated by the presence of *pals-5p*:*GFP* positive individuals. The infection in these populations was initiated by inoculating OrV on OP50-seeded plates containing axenic embryos. To maintain these infected populations, a 7 mm x 7 mm agar piece was chunked to fresh OP50-seeded plates. This transfer procedure was performed 2–3 times before starting experiments. From these plates, we selected 10 random L4 stage animals and placed them on plates seeded with either a natural bacterial strain or OP50 as a control. After a 96-hour period, we assayed infection in the offspring of these 10 animals, designated as Generation 1, by examining 100 young adults. We then repeated the process, picking another 10 random L4 larvae from Generation 1 and moving them to new plates seeded with the same bacteria as their respective parents. After 96 h, we assessed the Generation 2 offspring for infection, again examining 100 young adults.

Fig 6 illustrates this experiment, assessing infection via the *pals-5p*::*GFP* reporter and FISH staining of vRNA. S6B Fig replicates the experiment but relies solely on the *pals-5p*::*GFP* reporter for evaluating infection. S6C Fig shows a variation of the experiment where, instead of transferring 10 L4 animals, a 7 mm x 7 mm agar piece containing a random assortment of animals at various stages was transferred to new plates, and only the *pals-5p*::*GFP* reporter was assayed in Generation 2.

### Co-incubation of bacterial culture and virus inoculum

In Fig 7A we mixed 1 mL of liquid bacterial culture (bacterial strain indicated in the *x* axis of the figure, culture prepared as described in "Bacterial maintenance") with 1 mL of virus inoculum (prepared as described in "Virus maintenance"). This 1:1 mixture was incubated for 24 h at 20°C. After the incubation time, the tubes were centrifuged for 10 minutes at 5000 rpm and the supernatant was collected and passed through a 0.22 µm filter. We used 120 µL of the filtered supernatant to inoculate OP50-seeded plates. Axenic embryos were placed onto these plates, incubated at 20°C, and *pals-5p*::*GFP* activation was measured 72 hpi.

### Common garden inoculation experiment

In Fig 7B, we placed axenic ERT54 embryos on 90-mm plates seeded with OP50. These plates were then inoculated with 300 µL of virus inoculum, and animals were then maintained at 20°C for 24 h to enable virion intake. After these 24 h, animals were collected in M9 buffer and washed three times in 15 mL of M9 buffer. Washed animals were then split and transferred to non-virus inoculated plates seeded with the different bacterial strains indicated on the *x*-axis. The infection status of the animals was evaluated 48 h after being transferred to the new plates.

### Phylogenetic analysis

The bacterial 16S rDNA sequences used in the phylogenetic analysis can be found in S1 File. A multiple sequence alignment was performed using the MAFFT v7 tool [57] of MPI Bioinformatics Toolkit [58]. Aligned sequences were used to infer the phylogenetic tree by maximum likelihood using the IQ-TREE web server [59]. The phylogenetic tree and the plot of the Fig 1 were generated using the *phylo4d* function from the "phylosignal" package version 1.3 [60]. The phylogenetic signal in the trait data was assessed using the *lipaMoran* function, also

available in the "phylosignal" package. To account for multiple testing, we adjusted the obtained *P*-values using the Benjamini-Hochberg method. The level of significance was set at $P < 0.01$.

## Statistical analysis

All statistical analyses were conducted using R version 3.6.1 within the Rstudio development environment version 1.3.1093. The level of significance was set at $P < 0.05$. The specific statistical methods applied to each figure are detailed in the legend of the corresponding figure.

The analysis of variance, Tukey or Dunnett's post hoc test, and letter-based grouping for multiple comparisons were performed using the following functions: (i) the *aov()* function from the "stats" package version 3.6.1 [61] was used for conducting the ANOVA (ii) the *TukeyHSD()* function from the 'stats' package was used for performing the Tukey post hoc test (iii) Dunnett's post hoc test was carried out using the *glht()* function from the "multcomp" package version 1.4–23 [62] (iv) the *multcompLetters4* function from the "multcompView" package version 0.1.8 [63] was used for generating letter-based grouping for multiple comparisons. General linear models were performed using the *glm* function from the "stats" package. To account for the block effect, we employed linear mixed-effects or general linear mixed-effects models. The models were fitted to the data using the *lmer* or the *glmer* function from the "lme4" package version 1.1–21 [64]. Post hoc pairwise comparisons were performed using the *emmeans* function from the "emmeans" package version 1.4.7 [65]. Tukey's method was used to adjust for multiple comparisons in the pairwise comparisons.

Graphs were generated using the "ggpubr" package version 0.2.4 [66] in R.

## Supporting information

**S1 Fig. Proportion of animals with activation of the *pals-5p::GFP* reporter after Orsay virus inoculation on different bacteria.** Original data used in Fig 1, including here the experimental block structure. For each bacterial environment, three replicates of ca. 100 ERT54 animals were challenged with the OrV JUv1580. The proportion of animals activating the *pals-5p::GFP* reporter was measured 72 hpi. Data are presented as mean ± standard error. Letters over the bars indicate letter-based grouping for multiple comparisons. The yellow bar indicates *E. coli* OP50.
(TIF)

**S2 Fig. Fluorescence microscopy of ERT54 *[pals-5p::GFP; myo-2p::mCherry]* animals that were mock- or virus-inoculated in different bacterial environments.** Animals mock-inoculated with M9 or inoculated with OrV JUv1580 were visualized at 72 hpi using a 10x objective mounted on AxioImager M1 (Zeiss) compound microscope and images were captured with a PIXIS 1024 (Princeton instruments) camera. Scale bar represents 100 μm. GFP-positive animals can be seen in the intestinal cells of virus-inoculated animals on the three bacteria on top. Note that a less intense fluorescence can be seen in the posterior intestinal cells of animals in both mock- and virus-inoculated populations, especially in JUb44 environments. This posterior intestinal fluorescence is also observed in uninfected animals grown in OP50 [39]. We hypothesize that this fluorescence is caused by the 3'UTR of *unc-54* used in plasmids that generated the ERT54 strain. This DNA fragment probably contains cis-regulatory sites for the adjacent gene, *aex-5*, expressed in posterior intestinal cells as mentioned in [67].
(TIF)

**S3 Fig. Replication of some experiments shown in Fig 1. (A)** The impact of bacterial strains on viral infection of ERT54 nematodes is consistent across experimental blocks (performed on

different days). (**B**) New experiment with the bacteria that enhanced infection on the initial screening shown in Fig 1. Each data point represents a biological replicate, with 100 animals assayed per population. Data are presented as mean ± standard error. Asterisks on the graphs represent values of significance: *** $P < 0.001$; ** $P < 0.01$; * $0.01 < P < 0.05$; $P$ values higher than 0.05 are not labeled. Significance was calculated using a general linear model with bacteria as a factor and Dunnett's contrasts to compare all conditions against the *Escherichia* OP50 reference.
(TIF)

**S4 Fig. Test of several strains of selected bacterial species.** (**A**) Activation of the *pals-5p*::*GFP* reporter upon viral infection of ERT54 animals on different strains of the same bacterial species. (**B**) Activation of *pals-5p*::*GFP* reporter upon viral infection of animals on different *E. coli* strains. Each data point represents a biological replicate, with 100 animals assayed per population. Data are presented as mean ± standard error. Asterisks on the graphs represent values of significance: *** $P < 0.001$; ** $P < 0.01$; * $0.01 < P < 0.05$; $P$ values higher than 0.05 are not labeled. Significance was calculated using a general linear model with bacteria as a factor and Dunnett's contrasts to compare all conditions against the *Escherichia* OP50 reference.
(TIF)

**S5 Fig. Developmental growth rate of *C. elegans* on mixed bacterial environments.** Arrested axenic L1 larvae of the ERT54 strain were exposed to each monobacterial environment or to mixed environments composed of 20% of BIGb0102 and 80% of another bacterium (indicated in the *x* axis). The proportion of adults of 3 independent populations (100 animals per population) per environmental condition was observed after 46 and 62 h.
(TIF)

**S6 Fig. New experiments to confirm results.** (**A**) Activation of the *pals-5p*::*GFP* reporter upon OrV infection of animals on different single CeMbio strains and the whole CeMbio community. (**B**) Repetition of experiment shown in Fig 6, but only evaluating the activation of the *pals-5p*::*GFP* reporter. (**C**) Repetition of experiment shown in Fig 6, using a transfer of an agar chunk after 2 generations (detailed in methods). (**D**) Repetition of experiments, testing *drh-1* animals, shown in Fig 8D. (**E**) Susceptibility to viral infection on *Lelliottia* JUb276 of animals carrying a natural deletion allele of *drh-1*. Data are presented as mean ± standard error. Black symbols indicate the significance of the difference between the labeled bacteria and the *Escherichia* OP50 reference, while red symbols indicate the significance of differences between genotypes: *** $P < 0.001$; ** $P < 0.01$; * $0.01 < P < 0.05$; $P$ values higher than 0.05 are labeled as "ns". Significance was calculated using an analysis of variance with bacteria as a factor (Panels A,C,D), a general linear model where the factors were bacteria and generations (Panel D), or a general linear model where the factors were bacteria and host genotype (Panel E). In both cases Tukey contrasts were used for post hoc analyses.
(TIF)

**S7 Fig. Natural bacteria and anti-bacterial immune pathways.** (**A**) Experiment testing, on *Escherichia* OP50 or *Lelliottia* JUb276, virus susceptibility of mutants with alterations in different genes involved in the response against bacterial infections. We tested three biological replicates per genotype, with 100 animals assayed per population. Data are presented as mean ± standard error. (**B**) Experiment testing, on *Escherichia* OP50 or *Lelliottia* JUb276, virus susceptibility of *tir-1* mutants. We tested three biological replicates per genotype, with 100 animals assayed per population. Data are presented as mean ± standard error. (**C**) Fluorescence microscopy of the *sysm-1p*::*GFP* reporter. JU4289 animals carry the *agsl219[sysm-1p::GFP + ttx-3p::GFP]* transgene. Three day-old JU4289 animals grown at 20˚C in the indicated

bacterial environment were observed using a 10x objective on an AxioImager M1 (Zeiss) compound microscope. The Cherry and GFP fluorescence channels were captured with a PIXIS 1024 (Princeton instruments) camera and overlaid. PA14 and DB11 bacterial environments serve as positive control for the reporter activation. Scale bar represents 100 μm.
(TIF)

**S1 Table. List of bacterial strains used in this work.** The *pals-5p*::*GFP* column indicates that *C. elegans* is able to activate the *pals-5p*::*GFP* reporter on the tested strains after heat shock. For this test, 48 hours-old ERT54 animals were placed at 30˚C for 24 h, being observed right after the 24 h heat shock. FR is an abbreviation for France, GER for Germany, and USA for United States of America.
(DOCX)

**S2 Table. Nematode strains used in this study.**
(DOCX)

**S1 File. 16S rDNA sequences used for the phylogenetic analysis of the natural bacteria.**
(TXT)

**S2 File. Data generated in this study.**
(XLSX)

## Acknowledgments

We thank Aurélien Richaud for excellent technical assistance and advice and Tony Bélicard for the JU2624 strain. We wish to thank Emily Troemel for sharing the *zip-1(jy13)* strain. Some strains were provided by the CGC, which is funded by NIH Office of Research Infrastructure Programs (P40 OD010440). We thank the CNRS Segicel platform for generating the *drh-1(mcp553)* strain. We thank WormBase. We also thank members of Marie-Anne Félix, Marie Gendrel, and Henrique Teotónio's teams for fruitful discussions.

## Author Contributions

**Conceptualization:** Rubén González, Marie-Anne Félix.

**Data curation:** Rubén González.

**Formal analysis:** Rubén González.

**Funding acquisition:** Rubén González, Marie-Anne Félix.

**Investigation:** Rubén González.

**Project administration:** Rubén González, Marie-Anne Félix.

**Resources:** Marie-Anne Félix.

**Supervision:** Marie-Anne Félix.

**Visualization:** Rubén González.

**Writing – original draft:** Rubén González, Marie-Anne Félix.

**Writing – review & editing:** Rubén González, Marie-Anne Félix.

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
