## [Decision Letter · Decision Letter 0]

15 Aug 2023

Dear González,

Thank you very much for submitting your manuscript "Natural monobacterial environments modulate viral infection in Caenorhabditis elegans" for consideration at PLOS Pathogens. As with all papers reviewed by the journal, your manuscript was reviewed by members of the editorial board and by several independent reviewers. In light of the reviews (below this email), we would like to invite the resubmission of a significantly-revised version that takes into account the reviewers' comments. As you can see, several reviewers requested more information about the bacteria tested in this study. In particular it would be important to see more information about how these bacteria affect ingestion of virus in C. elegans, as effects here could have a significant impact on infection. Also important would be to show more quantitative and direct measurements of viral infection, particularly in places where conclusions are based on GFP reporter expression only, as an indirect read-out of infection.

We cannot make any decision about publication until we have seen the revised manuscript and your response to the reviewers' comments. Your revised manuscript is also likely to be sent to reviewers for further evaluation.

Sincerely,

Emily R. Troemel

Academic Editor

PLOS Pathogens

Sara Cherry

Section Editor

PLOS Pathogens

Kasturi Haldar

Editor-in-Chief

PLOS Pathogens

orcid.org/0000-0001-5065-158X

Michael Malim

Editor-in-Chief

PLOS Pathogens

orcid.org/0000-0002-7699-2064

Reviewer's Responses to Questions

**Part I - Summary**

Reviewer #1: This exciting study investigates the effect of different bacterial environments on the susceptibility of C. elegans to viral infection. The model system used is the popular model nematode C. elegans and its’ natural viral pathogen Orsay virus, an RNA virus that infects the worm intestine. The question of how microbes can influence viral infection is an important one, with broad relevance.

The authors address this question by conducting a targeted screen to test the susceptibility of C. elegans raised on 67 different bacterial monocultures to Orsay virus infection. They choose bacteria to test that were isolated from the natural environment of C. elegans, increasing the relevance of the study to the host-microbiome field. Their experimental design is rigorous, including repetitions of the initial infection reporter-based screen as well as follow-up testing on hits using RNA FISH to directly assess rates of viral infection. They identify several bacterial strains that enhance C. elegans susceptibility to Orsay infection, as well as a larger number of strains that decrease susceptibility. These effects are consistent across two different Orsay isolates with different infectivity rates.

In order to test the hypothesis that the enhanced susceptibility to infection is due to some bacterial strains providing inadequate nutrition or otherwise weakening the worms, they assessed the progeny production of worms raised on different bacteria. They concluded that there was no correlation between the effect of bacteria on progeny production and on viral susceptibility. While I agree that the data overall do not suggest general ill-health as the most likely explanation for the viral susceptibility effects seen with some of the bacterial strains, I think that this section could benefit from some additional clarifying language, as detailed below under Minor Revisions.

The authors next examined the effect of the bacterial environment on the horizontal transfer of the viral infection between worms. They were able to show that when OrV infected worms were transferred to infection-suppressing bacterial strains after an initial infection on OP50, the activation of the infection reporter was reduced in the 1st generation on the new bacteria, and almost completely eliminated by the 2nd. Although this data does in general support a reduction in horizontal transfer of infection, it is my opinion that the exclusive reliance on the infection reporter in this series of experiments is a significant weakness that undermines the conclusions in this section (see comment under Major Revisions).

The authors investigated the hypothesis that the infection-suppressing bacteria were somehow altering the infectivity of the virions through a two-pronged approach: pre-incubating the virus with the bacteria before infecting worms on OP50, and pre-infecting worms on OP50 then transferring to the infection-suppressing bacteria. These experiments convincingly demonstrated that the bacteria are exerting their infection-suppressing effects by altering host physiology, not virion infectivity. They further showed that heat-killed bacteria lost their infection suppression effect, lending further support to this hypothesis. They followed up with a series of bacterial mixing experiments, where they showed that the infection suppressing effect prevailed in a mixed bacterial population. Altogether, this data supports their conclusions that the infection-modulating effects of the bacteria are unlikely to be based on the nutritional content of the bacteria or bacterial effects on virions directly.

Finally, the authors also looked for interactions between the bacterial effects on OrV infection rates and host pathways known to be involved in resistance to viral infection. This series of experiments used both infection reporters and RNA FISH together with mutants to show that the bacterial effects on viral infection rates were, with a few exceptions, independent of the DRH-1, RDE-1, and CDE-1 pathways. The resistance to viral infection induced by Pseudomonas BIGb0477, Acinetobacter MYb10, and Lelliottia JUb276 were all found to be partially dependent on DRH-1. The complexities of the partial dependence/independence of each tested strain on the known pathways are discussed with an appropriate degree of nuance, and the conclusions add significantly to our knowledge of bacteria/host/virus interactions in C. elegans.

Overall this is a well-designed study that addresses an important topic in host-pathogen interactions, and lays the groundwork for future investigations of the immune enhancing/suppressing mechanisms of different bacteria. It will be of broad interest not only to the large community of C. elegans researchers, but also to the wider fields of host-pathogen and microbiome interactions.

Reviewer #2: In Gonzalez and Felix, the authors conduct a novel screen of 67 different bacterial strains naturally associated with C. elegans to determine if they influence infection by Orsay virus (OV), a natural viral pathogen of C. elegans. The authors identify both bacterial strains that enhance OV infection as well as bacterial species that promote resistance to OV infection. The authors demonstrate that prior incubation of OV virions with bacterial cultures that suppress OV infection does not reduce their ability to infect worms cultured on OP50 (E. coli) lawns, which are the common laboratory “food” used in C. elegans culture. This observation raises the fascinating possibility that these bacteria induce an antiviral response in worms. Interestingly, the authors data suggest that some of these “resistance-inducing” bacterial strains (e.g. BIGb0477, Myb10, and JUb276) may partially suppress OV infection by activation of responses that require DRH-1, which is involved in both antiviral RNAi responses and in inducing the Intracellular Pathogen Response (IPR). Remarkably, inactivation of RDE-1 (required for antiviral RNAi) or ZIP-1 (required for the IPR) does not reduce the ability of these resistance inducing bacteria to suppress OV infection, suggesting that these bacteria trigger a DRH-1-dependent mechanism to inhibit OV that is independent of RNAi or the IPR. Although this study does not provide much mechanistic depth in terms of understanding how bacteria promote or inhibit OV infection, there is significant novelty in this study in that it is the first to investigate bacteria-virus interactions in the C. elegans model on a large scale and is an important first step in establishing the worm model for studying such pathogen-pathogen-host interactions.

Enthusiasm for the manuscript was reduced by the poor organization of the manuscript (e.g. many figures are difficult to read/interpret). Also, there are often key experimental details left out that make it difficult to interpret some of the experiments presented (see specific comments below). A major concern is that the authors have not considered that some of the bacterial strains that are scored as inducing resistance to OV infection may in fact be promoting avoidance behavior by C. elegans, and thus reduced feeding which would in turn reduce ingestion of OV particles by worms. This point should be experimentally addressed. In addition, more quantitative methods should be used for assessing OV replication and not just infection rates, including qPCR-based methods to measure viral loads. The authors use GFP-based reporter strains and FISH staining to quantify infection rates but do not provide convincing evidence that the key bacterial strains highlighted from their initial screen actually increase or decrease OV replication.

Reviewer #3: See attached PDF for a formatted document.

Summary.

Host-microbial interactions can modulate host susceptibility to viral infections, but how this occurs is poorly understood. In the current study, the authors screen 67 natural bacterial strains for changes in Caenorhabditis elegans response to Orsay viral (OV) infection. Monocultures of most bacterial isolates reduce viral infection, which the authors claim is not due to viral degradation or poor nutrition. Surprisingly, C. elegans mutations known to compromise host antiviral responses are dispensable for the increased resistance.

Overall analysis.

Unraveling the complex interactions between host, viruses, nutrition, and microbiomes is critical for the development of novel treatments to improve human health. C. elegans is a powerful system to reveal these interactions; the authors should be commended for undertaking a large scale analysis of a diverse set of natural bacterial strains. However, there are several major concerns that severely limit enthusiasm for publishing the manuscript in the current form. First, the study is completely descriptive and provides little to no biological insight. There are interesting observations with insufficient follow-up; for instance, titrating in small amounts of natural bacteria acts dominantly to suppress pals-5p::GFP induction (a readout of host response to OV). Second, as all of the genetic analysis produced negative results, no putative mechanism is provided for why any bacterial food source would alter C. elegans host response to OV infection. However, intuitively the manuscript may be one discovery away from providing the necessary mechanistic insight necessary to warrant further consideration for PLOS Pathogens (suggestions below). Third, some of the experimental approaches are superficial and require additional rigor. For example, throughout the authors use a population assay for activation of pals-5p::GFP expression as a binary readout, which may lack sufficient nuance to adequately reveal relevant biology. Additional major but more easily addressable concerns include: 1. the authors overstate several conclusions based on indirect evidence, make a jump in logic, or commit a logical fallacy. Threads started within several supplemental figures need additional rigor and moved into the main text. Lastly, details are missing or difficult to infer in places and the quality of parts of several figures (embedded text) is poor. These and other concerns are described in additional detail below.

**Part II – Major Issues: Key Experiments Required for Acceptance**

Reviewer #1: The conclusion in line 158 “Natural bacteria eliminate OrV in infected nematode populations within two generations” is not fully supported by the evidence presented. This conclusion is drawn from an experiment in which worms are infected with OrV on OP50, then transferred to an infection-suppressing bacterial environment and allowed to reproduce. My issue with this conclusion is that the only readout used to assess infection in this experiment was pals-5p::GFP infection reporter readout, which is not a direct readout of viral infection but of worm immune response. The claim of “virus elimination” (lines 158 & 169-170) is therefore not fully supported by this data, which could also be explained by a change in the transcriptional activation of pals-5 in the progeny. The authors should either moderate their language on this point (in the lines noted above), or conduct RNA FISH staining to directly assess viral infection in order to support this claim.

Reviewer #2: The authors should examine if the key (5) bacterial strains they test throughout their study alter C. elegans feeding/avoidance behaviors.

Related to the first point, the authors should address the possibility that the key bacterial strains they investigate negatively impact C. elegans development (compared to OP50). Given that worms at different stages of development may have altered OV susceptibilities, the apparent impact of a bacterial culture on OV infection rates may be an indirect effect of altered growth/development rates. Reduced development rates caused by bacteria could also explain the reduced C. elegans brood sizes observed with most of these bacterial cultures.

The authors should validate key findings with more quantitative methods to assess differences in OV replication (e.g. qPCR of viral RNA to measure viral loads).

Reviewer #3: See attached for formatted document.

Major Concerns/Considerations.

1. The study is completely descriptive, which diminishes impact because insufficient biological insight is provided and there is no mechanism. Many paragraphs are essentially lists describing how each phenotype is affected by a particular natural bacteria.

2. Analysis is superficial in places. For example, while a 16S phylogenic analysis organizes the natural bacterial strains and reveals some clusters of bacterial strains that share a similar phenotype (e.g., Comamonas result in greater pals-5::GFP expression within a population), there is insufficient follow-up into what this means. For example:

a. Are any of the bacteria natural C. elegans pathogens (e.g., I thought Pseudomonas is a pathogen)? Do any limit survival/lifespan when maintained?

b. Which are gram positive or negative?

c. Are the bacteria rods or spheres?

d. How big are the bacteria (can they pass through the pharyngeal grinder)?

e. Do they colonize the intestinal lumen (opportunistic pathogen)?

f. What is the nutritional content of the bacteria? For example, some gram positive bacteria have thick cell wall polysaccharides covalently bound to peptidoglycan, which would increase relative sugar content.

3. There are several interesting observations that should inform a testable hypothesis but are not adequately explored or developed. Some examples of where the authors could focus in greater depth:

a. Growth on a natural bacterial results in loss of pals-5::GFP induction after OV infection but not after heat stress. This seems like a key observation! What does this mean? A trivial explanation would be that OV fails to adequately enter the host, which could be more rigorously tested using existing transgenic C. elegans strains that have the OV genome integrated under the control of an inducible reporter. Assuming OV entry is not impaired this suggests a specific adaptive response.

b. Titrating in small amounts of natural bacteria suppresses pals-5p::GFP induction and requires live-bacteria, which is somewhat surprising as the chosen bacteria are phylogenetically diverse. Nevertheless, this suggests a dominant effect that deserves additional consideration. Do some clades of natural bacteria secrete a dominant factor (which could be heat-labile) while others require ingestion? What happens if you mix natural bacteria that either enhance or limit pals-5::GFP expression? If you transfer worms grown in the presence of a natural bacteria, can it horizontally transfer resistance (suggests a secreted response between animals). I’m not sure of the best course, but the dominant effect seems like a key observation.

4. The authors use a population assay for pals-5::GFP expression as a binary readout, which may lack sufficient nuance to adequately reveal relevant biology and is too superficial. Additional level of secondary analysis is required (rigor). Does the absolute levels of GFP expression change? Which cells induce pals-5::GFP? Representative images must be included throughout. Throughout the authors describe the population assays as “similar levels of activation”, which is misleading.

5. The authors must more rigorously test whether a bacterial innate immune response is triggering an adaptive response to limit OV. Whether natural bacteria are inducing a bacterial innate immune response is underdeveloped and only superficially examined. Bacterial pathogens are diverse, induce specific reporters, and have unique signaling mechanisms (and genetic requirements).

6. Generally, animals are first introduced to natural bacteria at the same time OV is applied. The authors have not sufficiently distinguished whether exposure to a natural bacteria is inducing an acute adaptive response that is altering host-OV interaction, response, or dynamics of infection. Or whether a natural bacteria is generally protective. For example, if animals are maintained on a natural bacteria (that generates no bacterial innate immune response) as a food source for several generations, would they still be resistant to OV? If not, this suggests that switching food sources results in an adaptive adjustment period; perhaps a temporary alteration in metabolic flux as animals acclimate to the new food source. If animals are moved from a natural bacteria back to OP50, do animals retain resistance to OV? If so, for how many generations?

7. The authors describe horizontal transmission of OV (lines 158-163), but assess vertical transmission (Figure 3)? The text is misleading as horizontal expression isn’t actually tested. How far are into the infection are the L4 animals on OP50 just prior to infection? Were they pals-5::GFP positive? Were the animals chosen (in any capacity) for a similar intensity of GFP fluorescence (how is this normalized)? It is unclear from the schematic, exactly what the authors are doing and neither the figure legend nor method section contain the specific experimental details. This is problematic throughout and other experiments lack adequate description of the methods: for example, line 148: total brood sizes. How was this measured? How many animals? Details are missing.

8. Supplemental Figure 4 should be in the main text and be expanded to include a more rigorous analysis. For example, do these bacterial strains affect the growth rate of C. elegans, especially for the bacterial strains that repress pals-5::GFP induction? If the proportion of animals with pals-5 induction as readout, the authors need to confirm that animals fed with different bacterial are exactly at the same stage when infected.

9. It is unclear whether there are acute behavior changes after exposing animals to a new natural bacteria. Does pumping rate change immediately upon transfer to a new food source? Does feeding behavior change? Do animals avoid the bacterial lawn? It has previously been shown that some mutant C. elegans with apparent improved bacterial innate immunity were actually better able to sense and avoid the bacterial pathogen; these strains failed to show improved survival when a pathogenic bacteria lawn was spread across the entire plate. Did the authors uniformly coat the plate each bacteria? And was OV also distributed across the full plate? If this was not considered, then reduced pals-5::GFP expression within a C. elegans population after exposure to a mildly pathogenic bacteria could be explained trivially due to avoiding the lawn. The authors find that animals with reduced pals-5::GFP expression after grown on some natural bacteria results in extended reproductive spans, which would be consistent with this possibility. Note, the authors do find one natural bacteria that increases pals-5::GFP expression also increases reproductive span; this does not reject the hypothesis that animals with lower pals-5::GFP are avoiding another natural bacteria.

10. The authors over-interpret their results to draw conclusions that are too strong. Experiments do not always test what the authors conclude, and the authors tend to draw conclusions from indirect observations. For example, lines 135-138: “In conclusion, the Acinetobacter BIGb0102 environment enhances infection…” is an overstatement without quantification of endogenous pals-5 and RNA1/2 levels (RT-qPCR for each). Again, lines 154-155: “We thus conclude that the bacterial environments that enable strong viral infection did not do so by generally weakening the host.” is an overstatement as it is unknown whether viral infection is occurring at the same levels and “weakening” lacks informational content.

Other examples were also found and the authors should be more conservative in their conclusions. For example, line 184: “… can alter transcriptional response…” is a jump in logic that is not been tested experimentally. Line 208 subheading: “Unknown antiviral pathways are involved…” and line 290-91 “…unknown antiviral mechanisms…” are not supported with experimental results and is an “Appeal to Ignorance” logical fallacy (i.e., the absence of evidence is not evidence of absence).

11. Overall, the writing needs to be improved. Examples:

a. Some of the phrasing is awkwardly constructed: for example, line 32: “plays a key role in shaping various of its traits”.

b. In places the authors need to be specific, e.g., line 145: “to those suppressing it”, line 267: “certain bacterial environments” are vague and needs better clarity.

c. In other places the incorrect tense is used.

d. The authors should not refer to their prior work in 3rd person (e.g., line 141: Reported by Frezal).

e. There is inconsistent application of the same words (e.g., lines 259-260) strains of bacteria, natural strains of worms.

12. Figure legends for Figures 3-6 fail to describe number of tested animals, trials, statistical analysis, and significance. Links to primary data tables (supplementary files) are missing. Figure legend 2e indicates that 100 animals were assessed for total number of viable progeny with each column? Is this correct?

13. Line 272-275: “In C. elegans’ bacterial environments there is no distinction between food (nutrition) and biotic environment. This overlap is likely significant, given that the lipid content of the nematodes plays a crucial role in viral infections”. I absolutely agree! The authors should assess whether any of the bacterial clades alter major lipid stores, many straightforward assays are routinely used.

14. Heat killed bacteria are not a good food source and could confound results. UV-killed are a good alternative (or perhaps treated with antibiotics in some instances) to confirm results.

15. Was bacterial density normalized between natural bacterial strains? How well do each grow? Many bacteria do not grow as well in LB as E. coli.

16. In many figures the text has been rendered in a manner that makes it illegible.

17. Figure 6g is not mentioned in the text?

**Part III – Minor Issues: Editorial and Data Presentation Modifications**

Reviewer #1: Regarding lines 144-155: It would be beneficial here to clarify that progeny production is only one dimension of worm health; there are known instances of trade-offs between progeny production and other health measures such as lifespan or immunity. It would also help the clarity of this section to note that although the authors find brood size effects in Acinetobacter BIGb0102 (higher viral susceptibility but also higher brood size) and several of the other strains (lower viral susceptibility and also lower brood size) that seem to argue against the general-health hypothesis, they do find one strain that contradicts this pattern. Infection-enhancing strain Comamonas BIGb0172 also has lower brood size, and although they note that the brood size on BIGb0172 is similar to that on several other bacterial strains that protect against viral infection, this does not rule out a general health-suppressive effect at work in the specific case of BIGb0172.

There are issues with the spacing/fonts on nearly all of the figures that make some of the labels difficult to read in the PDF. This appears to be some sort of formatting/encoding issue.

The clarity of the figure legends could be improved in several places:

-In figure 2, the legend states that “Each data point represents an independent population of ~100 animals”; since individual data points are not shown in the graphs (only means), this is somewhat unclear. The authors should either adjust their language to make it clear whether they mean each experimental replicate used 100 worms (and how many replicates were performed), or alternatively show the individual data points for each replicate in addition to the mean.

-Overall, it is difficult to discern for most experiments shown (except figure 1) how many replicates were performed. This information should be added to the figure legends and the Materials and Methods.

In several places the text references an incorrect figure:

-on lines 204-205, it should be Supplementary figure 6D not 5D

-on line 232, it should be Supplementary figure 5C not 6D

-on line 233, there is no panel H in figure 6

-on line 242, there is no figure 7

-on line 254, it should be Supplementary Figure 6, not 7

Reviewer #2: 1. There is very little justification for the specific bacterial isolates used other than they are “natural bacterial strains”. Are certain strains/groups more commonly associated with C. elegans in the environment and thus may be more relevant to modulating virus susceptibility in the wild?

2. The authors should clearly state for each figure (e.g. in figure legends) which specific strain of C. elegans (e.g. ERT54?) was used for each assay.

3. Supplementary Figure 1-it would be helpful for the authors to comment further on how pals-5 activation was scored, including representative images of negative vs. positive activation phenotypes as well as control images where no virus was added because the authors indicate that the pals-5 reporter was not activated by any of the screened bacteria in the absence of virus infection but no data are shown to support this statement. Was GFP signal simply scored qualitatively as positive or negative if any part of the animal appeared green or was there some sort of quantitative measurement of GFP signal that had to meet a particular threshold to be scored as positive?

4. Supplementary Figure 2A- it is not clear why the y-axis is different between A and B. Are they both supposed to be percentage of animals showing pals-5 reporter activation? If so, why are the percentages so low in 2A (<2%) compared to 2B (~40-80%)?

5. I suggest re-reviewing the manuscript for grammatical errors and sentence structure throughout. For example, in the abstract line 17- naturally should be “natural” and in the Introduction line 33 “a key role in shaping various of its traits” should be re-worded.

6. Several of the figure labels are illegible (e.g. Figure 2A-C, Supplementary Figure 3A).

7. The authors identify a dramatic difference between Leucobacter luti BIG0106 and JUb18 strains relating to their effects on pals-5 reporter activation during OV infection but do not provide a possible explanation for this difference. The authors should ensure the identity of those bacterial strains are verified.

8. For the experiment in Figure 2D where two OV strains are compared, how did the authors ensure they were plating out similar doses of the two viruses? Also in Fig. 2, it is not clear what “Experiment 1 2 3” is referring to at the bottom.

9. In Fig. 2E, the authors argue that because one bacterial strain (BIGb0102) that enhanced OV infection rates leads to higher brood sizes that bacteria that enhance infection cannot be doing so by generally weakening host physiology. This cannot be concluded on the basis of this one example, because it is possible that some of the bacterial strains that enhance infection and that reduce worm brood sizes (e.g. BIGb0172) do so by reducing animal fitness. This relates to my earlier point regarding whether these different bacteria affect worm development rates compared to OP50 because altered development may lead to reduced brood sizes. Was brood size determined for each bacterial isolate in the absence of OV as well? Details regarding how brood size was measured should be included.

10. Line 232 refers to Supplementary Figure 6C but I think this should be 5C. Also line 245 refers to Supplementary Figure 7 but there is figure does not exist so I think this should be Supplementary Figure 6. In general, references to each figure in the text should be double-checked throughout.

11. The bars in Fig. 6G that are supposed to indicate WT or drh-1 genotypes look the same to me, precluding interpretation of these data.

Reviewer #3: See attached for formatted document.

1. Line 169: Figure 7B should be 3B. Line 242: 7G should be 6F. Line 245: Supplementary Figure 7 should be 6. The authors should double check Figure numbers throughout.

2. The definition of “natural” bacteria should be better defined. It seems odd to call E. coli “non-natural” bacteria (line 257).

3. Link to the full strain list (Table S2) needs to be mentioned in the first subsection of the Material and Methods.

4. Line 304: “It was created…” suggests a new strain was generated. Details on how to validate genotype (et cetera) are missing.

5. Line 340: “Evaluation of viral infection” should mention FISH.

6. S4 uses a colorblind unfriendly palette.

7. Fig. 1 use of terms enhancer and suppressor are not strictly known and inaccurate.

8. “Worms” is jargon and has an inherent negative connotation that diminishes the impact of C. elegans research; nematodes, C. elegans, or animals are better descriptors.

PLOS authors have the option to publish the peer review history of their article (what does this mean?). If published, this will include your full peer review and any attached files.

Reviewer #1: No

Reviewer #2: No

Reviewer #3: No
---

## [Decision Letter · Decision Letter 1]

4 Jan 2024

Dear Dr. Gonzalez,

We are pleased to inform you that your manuscript 'Naturally-associated bacteria modulate Orsay virus infection of Caenorhabditis elegans' has been provisionally accepted for publication in PLOS Pathogens.

Best regards,

Emily R. Troemel

Academic Editor

PLOS Pathogens

Ashley St. John

Section Editor

PLOS Pathogens

Kasturi Haldar

Editor-in-Chief

PLOS Pathogens

orcid.org/0000-0001-5065-158X

Michael Malim

Editor-in-Chief

PLOS Pathogens

orcid.org/0000-0002-7699-2064

Please address the minor comments from Reviewer #2, regarding labeling for timepoints on x-axis of Fig. 5A, and the Fig. S2A figure legend.

Reviewer Comments (if any, and for reference):

Reviewer's Responses to Questions

**Part I - Summary**

Reviewer #1: This exciting study investigates the effect of different bacterial environments on the susceptibility of C. elegans to viral infection. The model system used is the popular model nematode C. elegans and its’ natural viral pathogen Orsay virus, an RNA virus that infects the worm intestine. The question of how microbes can influence viral infection is an important one, with broad relevance.

The authors address this question by conducting a targeted screen to test the susceptibility of C. elegans raised on 67 different bacterial monocultures to Orsay virus infection. They choose bacteria to test that were isolated from the natural environment of C. elegans, increasing the relevance of the study to the host-microbiome field. Their experimental design is rigorous, including repetitions of the initial infection reporter-based screen as well as follow-up testing on hits using RNA FISH to directly assess rates of viral infection. They identify several bacterial strains that enhance C. elegans susceptibility to Orsay infection, as well as a larger number of strains that decrease susceptibility.

The authors have thoroughly addressed my major and minor concerns regarding their original submission. They have also added a substantial amount of new data that improves the paper, and have edited the text to significantly improve the clarity.

Overall this is a well-designed study that addresses an important topic in host-pathogen interactions, and lays the groundwork for future investigations of the immune enhancing/suppressing mechanisms of different bacteria. It will be of broad interest not only to the large community of C. elegans researchers, but also to the wider fields of host-pathogen and microbiome interactions.

Reviewer #2: In Gonzalez and Felix, the authors conduct a novel screen of ~70 bacterial strains naturally associated with C. elegans to determine if they influence infection by Orsay virus (OV), a natural viral pathogen of C. elegans. The authors identify both bacterial strains that enhance OV infection as well as bacterial species that promote resistance to OV infection. The authors demonstrate that prior incubation of OV virions with bacterial cultures that suppress OV infection does not reduce their ability to infect worms cultured on OP50 (E. coli) lawns, which are the common laboratory “food” used in C. elegans culture. This observation raises the interestingly possibility that these bacteria can modulate antiviral responses in worms. Although the authors do not provide a deep mechanistic dive into the nature of these responses, their data suggest that the repression of OV replication by some bacterial strains (JUb44 and BIG0172) is independent of DRH-1, a key regulator of known antiviral response pathways. Thus, these observations provide the groundwork for the exploration of additional mechanisms, independent of DRH-1 that may regulate innate antiviral responses in the worm. This manuscript will be of general interest to those studying microbe-microbe-host interactions.

In the revised manuscript, the authors have done a good job in taking an experiment-focused approach to addressing my comments and have improved the manuscript substantially. I only have a couple minor comments:

1. The labels for timepoints on the x-axis for Fig. 5A are not evenly spaced. It may be better to only indicate major ticks (time points) or define the x-axis as hours (h) and then remove the “h” after each time point to improve clarity of the labels.

2. Supplementary Fig. 2 legend- last sentence should be corrected to “this DNA fragment probably contains…”.

**Part II – Major Issues: Key Experiments Required for Acceptance**

Reviewer #1: (No Response)

Reviewer #2: My prior experimental suggestions have been adequately addressed in the revised manuscript.

**Part III – Minor Issues: Editorial and Data Presentation Modifications**

Reviewer #1: (No Response)

Reviewer #2: 1. The labels for timepoints on the x-axis for Fig. 5A are not evenly spaced. It may be better to only indicate major ticks (time points) or define the x-axis as hours (h) and then remove the “h” after each time point to improve clarity of the labels.

2. Supplementary Fig. 2 legend- last sentence should be corrected to “this DNA fragment probably contains…”.

PLOS authors have the option to publish the peer review history of their article (what does this mean?). If published, this will include your full peer review and any attached files.

Reviewer #1: No

Reviewer #2: No

---

## [Editor Report · Acceptance letter]

12 Jan 2024

Dear Dr. González,

We are delighted to inform you that your manuscript, "Naturally-associated bacteria modulate Orsay virus infection of Caenorhabditis elegans," has been formally accepted for publication in PLOS Pathogens.

Best regards,

Michael Malim

Editor-in-Chief

PLOS Pathogens

orcid.org/0000-0002-7699-2064